# Fully lignocellulose-based PET analogues for the circular economy

Xianyuan Wu [1], Maxim V. Galkin [1], Tobias Stern[2], Zhuohua Sun [3✉] & Katalin Barta [1,4✉]

Polyethylene terephthalate is one of the most abundantly used polymers, but also a significant pollutant in oceans. Due to growing environmental concerns, polyethylene terephthalate alternatives are highly sought after. Here we present readily recyclable polyethylene terephthalate analogues, made entirely from woody biomass. Central to the concept is a two-step noble metal free catalytic sequence (Cu20-PMO catalyzed reductive catalytic fractionation and Raney Ni mediated catalytic funneling) that allows for obtaining a single aliphatic diol 4-(3-hydroxypropyl) cyclohexan-1-ol in high isolated yield (11.7 wt% on lignin basis), as well as other product streams that are converted to fuels, achieving a total carbon yield of 29.5%. The diol 4-(3-hydroxypropyl) cyclohexan-1-ol is co-polymerized with methyl esters of terephthalic acid and furan dicarboxylic acid, both of which can be derived from the cellulose residues, to obtain polyesters with competitive Mw and thermal properties ($T_g$ of 70–90 °C). The polymers show excellent chemical recyclability in methanol and are thus promising candidates for the circular economy.

[1] Stratingh Institute for Chemistry, University of Groningen, Groningen, The Netherlands. [2] University of Graz, Institute of Systems Sciences, Innovation and Sustainability Research, Merangasse 18/I, 8010 Graz, Austria. [3] Beijing Key Laboratory of Lignocellulosic Chemistry, Beijing Forestry University, No. 35 Tsinghua East Road Haidian District, Beijing 100083, P. R. China. [4] Department of Chemistry, Organic and Bioorganic Chemistry, University of Graz, Heinrichstrasse 28/II, 8010 Graz, Austria. ✉email: sunzhuohua@bjfu.edu.cn; katalin.barta@uni-graz.at

With an annual production of 70 million tons globally, polyethylene terephthalate (PET) is one of the most widely used polymers worldwide, indispensable for the manufacturing of packaging material, clothing, fibers, and single-use beverage bottles[1,2]. However, its accumulation in landfills and oceans has been estimated to reach up to 530 million tons to date[2], which accounts for near-catastrophic environmental pollution[2–4]. Moreover, most of the PET is still, typically, produced from fossil resources by copolymerization of ethylene glycol (EG) and terephthalic acid (TPA)[5].

Thus, there is a tremendous incentive to obtain readily recyclable[6,7] or upcyclable[8], fully bio-based PET alternatives[9,10] in order to implement circular economy approaches[11–15]. This will require the development of robust catalytic methods and comprehensive biorefinery strategies[16–19].

A well-known emerging industrial approach is the replacement of petrol-based TPA with furan dicarboxylic acid (FDCA) from sugar-derived 5-hydroxymethylfurfural (5-HMF)[20,21]. Other laboratory-scale examples focus on pathways to source EG[22,23] and TPA from biomass[24–27]. Lignin-derived monomers, such as ferulic or syringic acids, have been investigated for the preparation of PET, PET mimics and PET reinforced plastics[8,9,28,29] (Fig. 1A). Beckham and coworkers developed a smart upcycling route where PET was modified by EG and muconic acid to give an unsaturated polyester, which was subjected to cross-linking to produce fiberglass-reinforced plastics[8].

Reductive catalytic fractionation (RCF)[30–33] has shown a powerful strategy for obtaining high yields of aromatic monomers from lignocellulose that can be converted to a variety of polymer building blocks including TPA[24] but also others such as 4-propylcyclohexanol[34,35], bisphenol 5,5-methylenebis(4-n-propylguaiacol)[36] and 3,3′-ethylenebis(4-n-propylsyringol)[37] for making diverse types of polymers. Epp's group synthesized high-performance adhesives from 4-n-propylsyringol and RCF mixtures[38].

Here we present a comprehensive biorefinery strategy for constructing PET analogues, as well as gasoline range and jet-range fuels, based entirely on woody biomass (Fig. 1B). In our unique approach, lignin gives the aliphatic diol building block, while cellulose may provide the necessary aromatic diacid components (FDCA or TPA) in the developed polyesters. Central to the method is the catalytic funneling[24,39–42] of native lignin by a non-noble metal two-step catalytic sequence, which results in 4-(3-hydroxypropyl) cyclohexan-1-ol (PC), obtained as a single compound in an excellent isolated yield of 11.7 wt% (on lignin basis), and other product streams are convertible to gasoline range cyclohexane derivatives and high energy density fuel range bicyclic alkane, achieving a total carbon yield of 29.5%. Notably, this funneling strategy allows overcoming tedious and expensive product isolation and purification protocols, providing a single polymer building block PC from complex biomass feed as well as other usable product streams. This leads to overall excellent lignocellulose utilization and the synthesis of fully wood-based polyesters. The polyesters poly (PC/TPA) and poly (PC/FDCA) display excellent thermal properties ($T_g$ = between 75–90 °C) that compare favorably to commercially available PET and are readily recyclable in methanol. Preliminary techno-economic analysis shows the feasibility of the developed process, thus overall these fully bio-based polyesters are promising candidates for the circular economy.

## Results

### Catalytic defunctionalization of lignin-derived platform chemicals.
We have previously reported on the RCF of various lignocellulose species with excellent selectivity to 4-n-propanolguaiacol (1 G) using pine and to a mixture of 4-n-propanolguaiacol (1 G) and 4-n-propanolsyringol (1 S) using poplar/beech lignocellulose, with Cu20-PMO as a catalyst[43]. The advantage of this method is that 1 G and 1 S feature an aliphatic alcohol moiety. Thus, first, we envisioned the catalytic funneling of these aromatic platform chemicals to the aliphatic diol PC to test its suitability for making PET analogues. This should be achieved by highly selective demethoxylation/hydrogenation, while maintaining this primary alcohol functionality; a challenging reaction, since the γ-alcohol may participate in direct hydrogenolysis or undergo a dehydrogenation/decarbonylation cascade, especially at higher temperatures[43]. In fact, such reactions have been typically investigated using simpler substrates or lignin-derived bio-oils, resulting in a higher degree of defunctionalization[44,45].

To circumvent the above-mentioned pathways, we first evaluated a range of commercially available heterogeneous catalysts (Fig. 2A and Supplementary Table 1) that may exhibit good reactivity in demethoxylation/hydrogenation under relatively mild conditions (100 °C, for 2 h, using 10 bar $H_2$) in isopropanol as a solvent, using 1 G as a substrate (Supplementary Note 2.1). Both Ni/γ-Al$_2$O$_3$ and Ni/C were inactive, and only moderate success was achieved with Ni/SiO$_2$-Al$_2$O$_3$. Noble metal catalysts demonstrated high 1 G conversions (83.4%, Pd/C and 100%, Ru/C), but low selectivity for the desired diol PC (11.5%, Pd/C and 29.9%, Ru/C), while displaying higher selectivity to 4-(3-hydroxypropyl)-2-methoxycyclohexanol (**1**) (51.6%, Ru/C, 30.5%, Pd/C) and other side products such as 4-ethylcyclohexanol (**2**), 4-propylcyclohexanol (**3**), 4-propylphenol (**2H**) and 4-(3-hydroxypropyl)phenol (**1H**).

Gratifyingly, using Raney Ni led to 84.8% 1 G conversion and 84.4% PC selectivity. Diol PC was obtained as a 1:2 mixture of cis: trans isomers (determined by GC-FID (Supplementary Fig. 28b) and $^1$H-NMR (Supplementary Fig. 32). The addition of $H_2$ gas appeared necessary to suppress the dehydrogenation/decarbonylation cascade otherwise leading to loss of selectivity (Fig. 2B and Supplementary Table 2). The reaction mixture obtained under optimized conditions consisted of compounds PC, **1**, and **2**.

Further evaluating a range of solvents using Raney Ni (10 bar $H_2$, at 100 °C), it was confirmed that isopropanol is a superior reaction medium (Fig. 2C and Supplementary Table 3), in line with excellent previous work of Rinaldi and coworkers[44] using this solvent/catalyst combination for transfer hydrogenation purposes. Thus, we attribute this higher rate of catalytic funneling towards PC to the high transfer hydrogenation activity of Raney Ni as well as its lower tendency for aromatic ring hydrogenation compared to noble metal catalysts[46] (Supplementary Note 5). Optimization led to an 84.8% yield of PC at 120 °C for 2 h, applying 10 bar $H_2$ (Fig. 2D and Supplementary Table 4).

Based on the product distribution, PC could have formed through demethoxylation, followed by hydrogenation or vice versa. To gain more insight into plausible reaction pathways from 1 G leading to PC, further kinetic studies were undertaken (Supplementary Notes 2.3, 2.4). Reaction intermediates 1H and **1** showed a significant difference in reactivity when subjected to optimized reaction conditions (Supplementary Tables 7, 8 and Fig. 2E). While **1** displayed very low conversion, 1H was readily hydrogenated to the desired product PC. As a comparison, the aromatic ring hydrogenation rate of 1H ($k_2 = 0.4400$ min$^{-1}$) was much higher than demethoxylation of **1** ($k_4 = 10^{-10}$ min$^{-1}$). Furthermore, the rate of hydrogenation of 1 G to **1** ($k_3 = 0.0017$ min$^{-1}$) and the rate of demethoxylation of 1 G to 1H ($k_1 = 0.0082$ min$^{-1}$) were comparable. Summing up these observations, the following order was observed: $k_2 > k_1 > k_3 > k_4$, where demethoxylation of 1 G to 1H is the rate-limiting step (Fig. 2E). Therefore, we assume that the catalytic conversion of 1 G

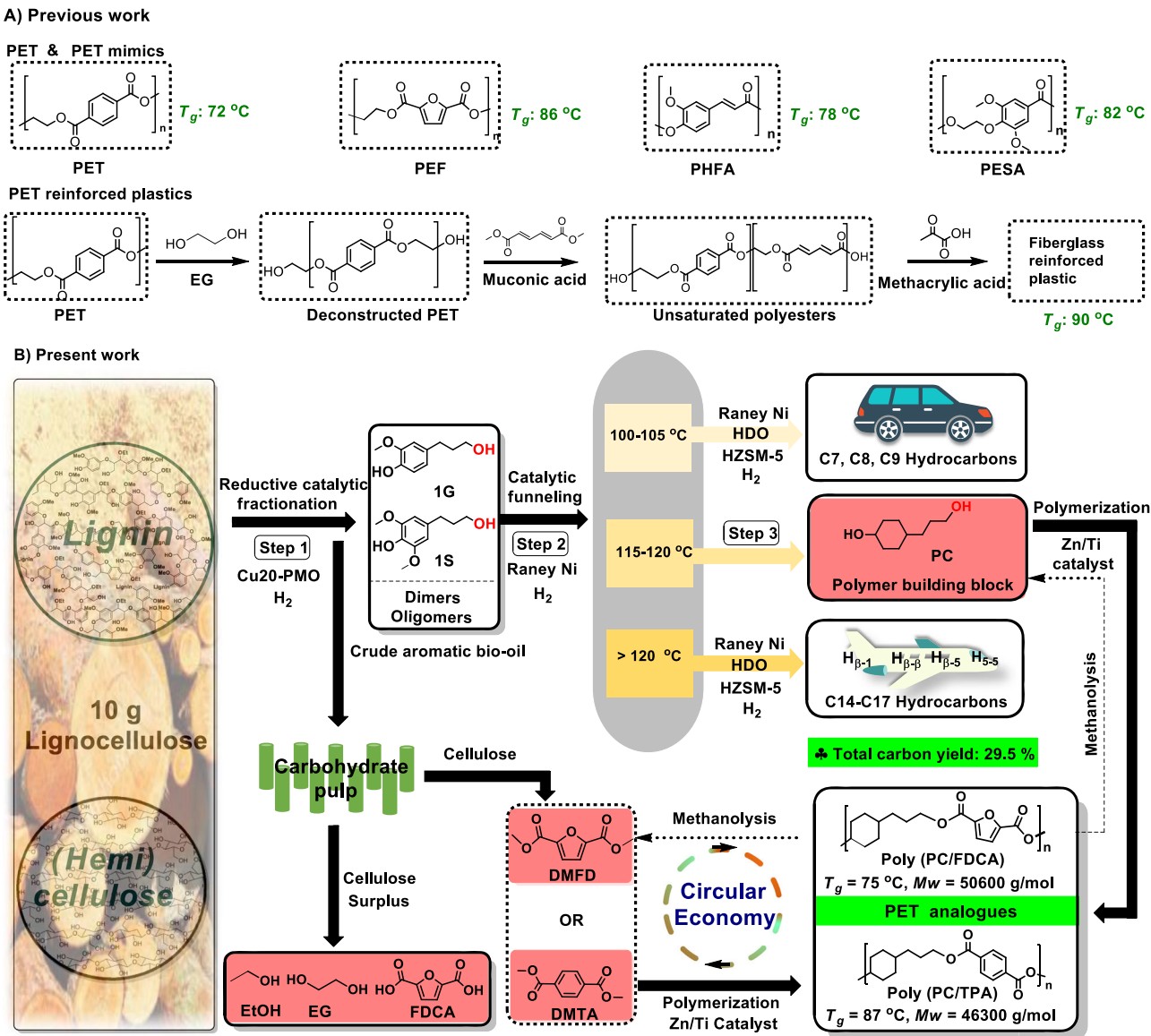

**Fig. 1 An overview of bio-based PET analogues derived from lignocellulose. A** Previous representative work to produce bio-based PET, PET mimics, and PET reinforced plastics. **B** Our comprehensive biorefinery strategy to produce fully lignocellulose-based, recyclable PET analogs and other valuable products following three key steps: (1) Reductive catalytic fractionation (RCF) of lignocellulose over Cu20-PMO catalyst to give a crude lignin oil rich in 1 G and 1 S bearing a primary alcohol functionality; (2) Catalytic funneling of the ethyl-acetate extracted RCF mixtures to PC diol and other product streams using Raney Ni/isopropanol, (3) Copolymerization of PC with methyl esters of FDCA and TPA to give fully bio-based and recyclable polyesters poly (PC/TPA) and poly (PC/FDCA). Potential valorization of the carbohydrate residues obtained upon RCF of the same lignocellulose source to the aromatic diacids FDCA and TPA is necessary for the copolymerization, while any surplus of cellulose can be converted to bioethanol and/or ethylene glycol (EG) (see Supplementary Note 1 for more details).

proceeds through demethoxylation to give compound 1H, followed by its hydrogenation to PC, while hydrogenation of 1 G to intermediate **1** is considered a parallel side reaction, and other side reactions are relatively slow (Fig. 2F and Supplementary Fig. 27).

Next, 1 S comprising extra functionality was evaluated as substrate (Supplementary Table 5), using Raney Ni catalyst under previously optimized reaction conditions (140 °C, 20 bar, 2 h, isopropanol). Gratifyingly, PC was obtained in an 84.6% yield (Table 1). Moreover, in order to establish a catalytic funneling strategy, model mixtures of 1 G/1 S and 1 G/1 S/1H that are the main components of bio-oils derived by RCF of hardwood or switchgrasses, were successfully converted into PC with an excellent yield of 84.8 and 86.3% respectively.

**Catalytic funneling of crude RCF mixtures**. Next, we applied the developed Ni-based method for the chemo-catalytic funneling of the bio-oil obtained through RCF of beech lignocellulose (Supplementary Note 2.5). Treating 2 g of beech wood over Cu20-PMO using 40 bar $H_2$ at 180 °C gave crude aromatic bio-oil (Fig. 3) rich in the desired phenolic monomers 1 G (31.3 mg) and 1 S (65.2 mg). Further in-depth analysis by 2D-HSQC (Supplementary Fig. 30), GC-FID (Supplementary Fig. 28b), and GPC (Supplementary Fig. 29) revealed the presence of additional monomers 2 S (21.8 mg), 2 G (5.2 mg), 3 S (4.6 mg) as well as lignin dimers (Supplementary Fig. 28a), oligomers and sugar residues (Supplementary Figs. 29, 30). For comparison, we also performed the RCF step using Pd/C and achieved a high 1 G/1 S selectivity (Supplementary Table 10).

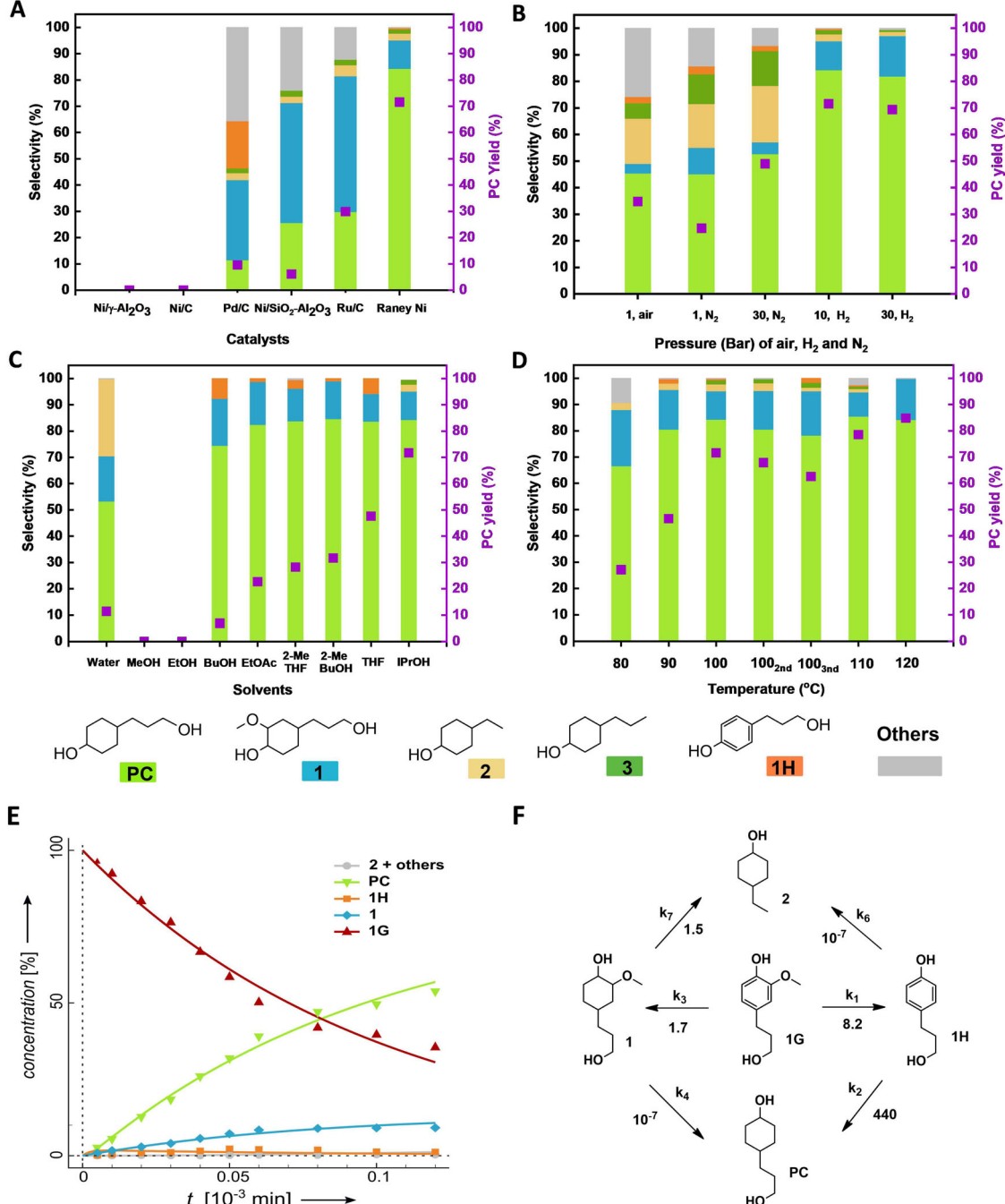

**Fig. 2 Establishing optimal reaction conditions for the catalytic conversion of 1 G to PC.** Standard reaction conditions unless otherwise specified: 1 G (1.1 mmol, 0.200 g), 1 g wet Raney Ni, 100 °C, 2 h, 10 bar $H_2$, 15 mL solvent, 10 mg dodecane. Screening of **A** commercially available heterogeneous metal catalysts; **B** different gas phases: air, $N_2$, and $H_2$; **C** various solvents; **D** reaction temperature; **A–D** For numerical values see Supplementary Supplementary Table S1–S4; Conversion and yield values were determined by GC-FID using calibration curves and internal standard; **E** Proposed reaction network and calculated apparent rate constants from data fitting, utilizing the DynaFit software ($k_{app}$ in $10^{-3}$ min$^{-1}$); **F** Fitting of the experimental data for catalytic conversion of 1 G, **1**, 1H, and PC over wet Raney Ni catalyst. Reaction conditions: 1.1 mmol substrates, 0.500 g wet Raney Ni, 100 °C, 20 bar $H_2$, 20 mL isopropanol, 20 mg dodecane as internal standard. Data were presented in detail in Supplementary Tables S6–S8.

Initial attempts to directly subject crude aromatic bio-oil from Cu20-PMO treatment, to further catalytic processing was unsuccessful, likely due to catalyst deactivation. Therefore, a simple fractionation protocol with EtOAc was implemented to get rid of residual lignin oligomers, sugars, and small amounts of organic acids, that may be detrimental to the catalysis, as earlier reported[47]. Gratifyingly, after this treatment, the—still multi-component—EtOAc extracts were smoothly converted to a

mixture of aliphatic alcohols rich in PC (52.4 mg, 74.2%) and **1** (16.3 mg) (Fig. 3), originating from 1 G and 1 S, which represents 13.9 wt% yield based on lignin, an efficiency of 56.3% given that theoretical maximum yield is 24.7 wt% (Supplementary Note 1). Furthermore, compounds 2 S, 2 G, 3 S, and 3 G also contained in EtOAc extracts in smaller amounts, were converted to alcohols **2** (5.7 mg) and **3** (11 mg), while aliphatic dimers and oligomers (totally approximately 40 mg) were also formed, and these

**Table 1 Catalytic conversion of model compounds 1 G, 1H, 1 S, and lignin oil to PC diol.**

| Entry | Substrates | T [°C] | T [h] | p [bar] | Conv. [%] | Selectivity[f] [%] | | | | | PC[e] Yield[f] [%] |
|---|---|---|---|---|---|---|---|---|---|---|---|
| | | | | | | PC | 1 | 2 | 3 | others | |
| 1 | 1 S[a] | 120 | 2 | 10 | 91.0 | 69.3 | 7.9 | 6.7 | 3.4 | 2.4 | 63.3 |
| 2 | 1 S | 130 | 2 | 20 | 98.1 | 83.3 | 10.6 | 1.9 | 1.6 | 1.3 | 81.7 |
| 3 | 1 S | 140 | 2 | 20 | 100 | 84.6 | 10.6 | 1.6 | 2.1 | 1.1 | 84.6 |
| 4 | 1 G/1 S[b] | 140 | 2 | 20 | 100 | 84.8 | 9.5 | 2.4 | 2.0 | 1.3 | 84.8 |
| 5 | 1 G/1 S/1H[c] | 140 | 2 | 20 | 100 | 86.3 | 8.9 | 2.3 | 1.6 | 0.9 | 86.3 |
| 6 | PC from pine[d] | 150 | 3 | 30 | 100 | 76.5 | 17.6 | - | - | 5.9 | 76.5 |
| 7 | PC from poplar[d] | 150 | 3 | 30 | 100 | 75.1 | 18.5 | - | - | 6.4 | 75.1 |
| 8 | PC from beech[d] | 150 | 3 | 30 | 100 | 74.2 | 19.2 | - | - | 6.6 | 74.2 |

General conditions: 1 g wet Raney Ni, 15 mL isopropanol, 10 bar $H_2$, 20 mg dodecane as an internal standard.
[a]1 mmol (0.212 g) 1 S.
[b]0.55 mmol (0.100 g) 1 G, 0.47 mmol (0.100 g) 1 S.
[c]0.27 mmol (0.050 g) 1 G, 0.23 mmol (0.050 g) 1 S, 0.65 mmol (0.100 g) 1H.
[d]2 g Lignocellulose.
[e]PC was obtained as a mixture of isomers (cis: trans = 1: 2).
[f]Selectivity and yield values were determined based on a molar basis using dodecane as internal standard (See Supplementary Note 1.3).

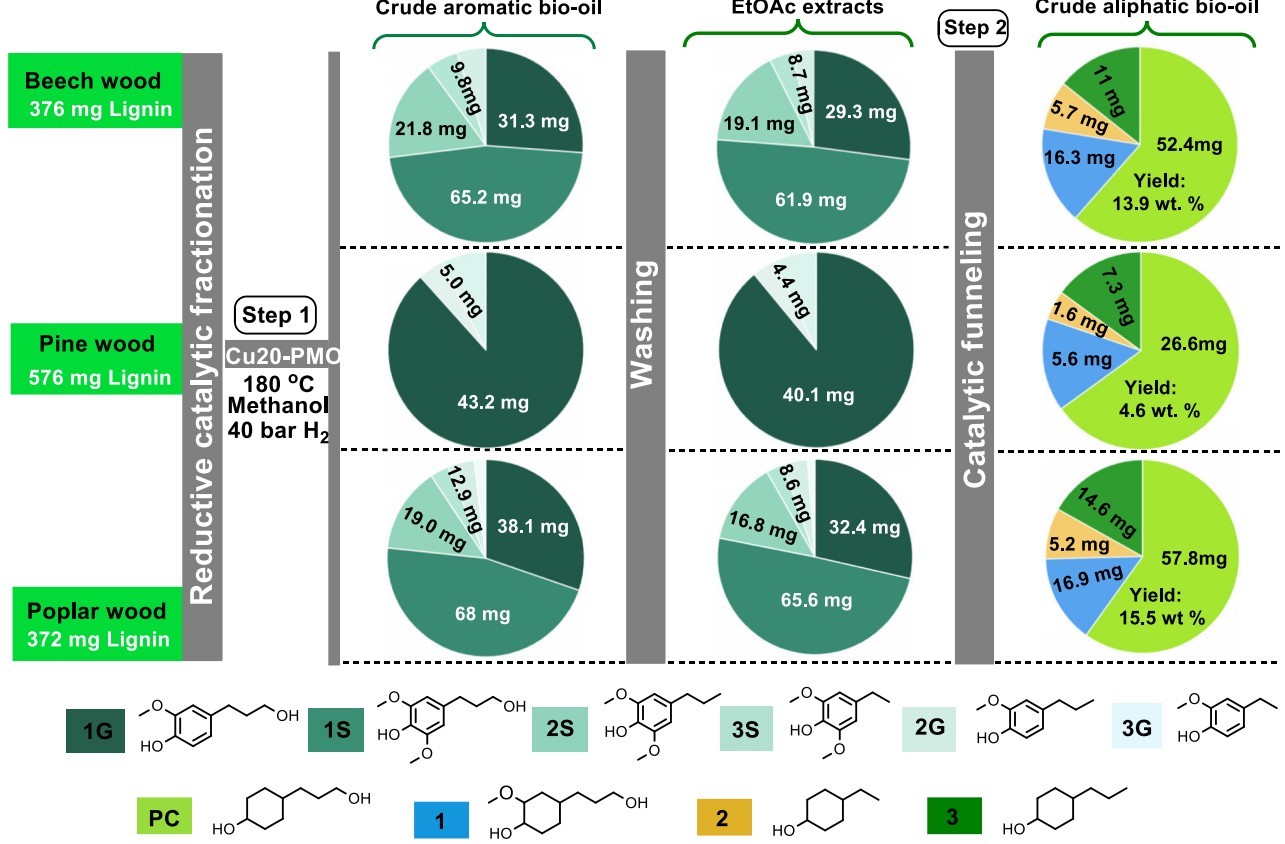

**Fig. 3 A catalytic reaction protocol toward the production of PC directly from a crude RCF mixture of beech, pine, and poplar lignocellulose.** Step 1: RCF of beech, pine, and poplar wood over Cu20-PMO catalyst to give crude aromatic bio-oil. The crude aromatic bio-oil was subjected to fractionation with EtOAc and washing with saturated $NaHCO_3$ and brine to give EtOAc extracts; Step 2: Catalytic funneling of EtOAc extracts to PC and other cyclohexane derivatives over Raney Ni/isopropanol. The yield was given based on lignin content. In this figure, only the volatile, monomeric products are displayed. More quantification, analysis, and characterization details can be found in Supplementary Note 2.5.

accounted for less than 28.3 wt% of the defunctionalized and hydrogenated product mixture (Supplementary Fig. 28a). PC was obtained by column chromatography from the aliphatic product mixture in an excellent 11.7 wt% isolated yield.

To evaluate the generality of the method, pine and poplar lignocellulose (2 g each) were also subjected to RCF to give crude aromatic bio-oil, resulting in 1 G from pine wood in 7.5 wt% yield

(88.0% selectivity), as well as a mixture of 1 G (30% selectivity) and 1 S (46.1% selectivity) from poplar wood in 28.2 wt% combined yield as summarized in Supplementary Table 9. After applying the developed fractionation protocol, catalytic funneling of these RCF oils gave 4.6 wt% yield to PC (76.5% selectivity) for pine and 15.5 wt% yield to PC (75.1% selectivity) for poplar lignocellulose (Table 1 - Entry 6, 7 and Fig. 3).

*Synthesis of PET analogues from lignin-derived PC.* Considering our strategy, where PC is obtained from lignin through RCF and catalytic funneling, a portion of the residual cellulose may be utilized for the production of FDCA or TPA by applying methods that are already established (Supplementary Note 1), thereby giving fully wood-based PET analogues. Therefore, we have selected methyl esters of aromatic diacids TPA and FDCA for copolymerization with PC (For details on polymer synthesis see Supplementary Note 3). Thus, the methyl ester of TPA was copolymerized with PC using Zn(OAc)$_2$[48] as a catalyst to result in poly (PC/TPA) in up to 75.0% yield (Table 2, Entry 1), while copolymerization of PC with the methyl ester of FDCA using Titanium (IV) butoxide (TBT)[49] as catalyst gave poly (PC/FDCA) in excellent, up to 92.3% yield (Table 2, Entry 5). Overall, the yields of the obtained polyesters, ranging from 50–95% (Table 2, Entries 1-11) largely depended on the type of catalysts, polymerization conditions, and the comonomer partners used. In addition, Sb$_2$O$_3$ was also tested and compared for the synthesis of poly (PC/TPA) but lower reactivity was found (Supplementary Table 11). All prepared polymers displayed a molecular weight range of 20–50 kg mol$^{-1}$ (Table 2). Selected samples of poly (PC/TPA) and poly (PC/FDCA) were also subjected to a purification protocol[48], which further enhanced their Mw to 50 kg mol$^{-1}$ (Table 2, Entries 3, 6, and Supplementary Figs. 56, 66).

*Structural characterization.* The obtained PET analogues were structurally characterized by $^1$H, $^{13}$C NMR, 2D NMR (Supplementary Figs. 36–46), and FT-IR (Supplementary Fig. 68) as extensively discussed in Supplementary Note 3.1 briefly below. Featuring both an aliphatic primary and secondary alcohol moiety, diol PC is asymmetric in nature and this leads to different reactivity and unit connectivity when forming an ester linkage with diacids at each end of the molecule, hence forming three distinct dyads, namely head-to-tail (H-T), head-to-head (H-H), and tail-to-tail (T-T) (Fig. 4A). We have first assigned the main-chain sequence of poly (PC/TPA), where PC is a mixture of cis and trans isomers (ratio of 1: 2), by $^1$H, $^{13}$C NMR, and 2D-HSQC spectroscopy (Fig. 4C) (for 2D HMBC see Supplementary Fig. 46). The polymer dyad structure was established by $^{13}$C NMR (Fig. 4B and Supplementary Fig. 45) analysis that displayed three distinguishable groups of carbonyl signals (C8). Based on related literature data, the high field signals of C8 at 165.24 ppm (C8 cis) and 165.42 ppm (C8 trans) were assigned to H-H type[50] and the low field C8 signals (cis and trans overlapped) at 166.05 ppm to T-T type structure[51], and the signals in-between (165.28, 165.48, and 165.97 ppm) to mixed (H-T, T-H) type connectivity units. The signal assignment was further confirmed by 2D $^1$H-$^{13}$C HMBC (Supplementary Fig. 46), where protons H1, correlating with C8 at 165.2–165.5 ppm and at 165.95–166.05 ppm, respectively displayed an H type bonding, while proton H7 correlating with C8 at 165.9–166.1 ppm was assigned to a T type bonding. The proposed structural assignment was further supported by comparing the spectral data of poly (PC/TPA) with poly (PC$_{trans}$/TPA) and poly (PC$_{cis}$/TPA), separately made from pure trans or cis PC (Supplementary Fig. 47), discussed in detail in the Supplementary Note 3.1. The signals of C8 were assigned according to T-T, T-H, and H-H types of dyads, which displayed a random distribution: [H-H] = [T-T] = 0.25 and [H-T] = 0.50, as quantified by integration of quantitative $^{13}$C NMR spectrum (Supplementary Fig. 45). The structural analysis of the polymer poly (PC/FDCA) is discussed in detail in Supplementary Note 3.2, similarly to the poly (PC/TPA), it is a random type polymer.

*Thermal analysis.* Overall, the favorable thermal properties of poly (PC/TPA) and poly (PC/FDCA) indicate the potential for the use of these polymers in applications similar to those of PET

**Table 2 Molecular weight distributions and thermal properties data for synthesized polymers.**

| Entry | Substrate | Catalyst | Time (h) | Yield[d] [%] | Mw[e] [g mol$^{-1}$] | Mn[e] [g mol$^{-1}$] | Đ[e] | T$_{5\%}$[f] [°C] | T$_{90\%}$[f] [°C] | T$_g$[g] [°C] |
|---|---|---|---|---|---|---|---|---|---|---|
| 1 | Poly (PC/TPA) | Zn(OAC)$_2$ | 1 | 75.2 | 22,900 | 11,700 | 1.94 | 320 | 373 | 76 |
| 2 | Poly (PC/TPA) | Zn(OAC)$_2$ | 3 | 72.1 | 31,700 | 14,100 | 2.24 | 329 | 381 | 81 |
| 3 | Poly (PC/TPA)[a] | Zn(OAC)$_2$ | 3 | 50.8 | 46,300 | 14,000 | 3.29 | 319 | 378 | 87 |
| 4 | Poly (PC/FDCA) | TBT | 1 | 88.0 | 33,300 | 15,600 | 2.13 | 280 | 347 | 73 |
| 5 | Poly (PC/FDCA) | TBT | 3 | 92.3 | 33,500 | 12,000 | 2.79 | 320 | 371 | 71 |
| 6 | Poly (PC/FDCA)[a] | TBT | 3 | 68.4 | 50,600 | 12,500 | 4.03 | 278 | 335 | 75 |
| 7 | Poly (PC$_{cis}$/TPA) | Zn(OAC)$_2$ | 1 | 84.0 | 21,200 | 9000 | 2.34 | 319 | 367 | 72 |
| 8 | Poly (PC$_{trans}$/TPA) | Zn(OAC)$_2$ | 1 | 53.2 | 16,800 | 8100 | 2.06 | 313 | 376 | 51 |
| 9 | Poly (PC/**1**/TPA)[b] | Zn(OAC)$_2$ | 1 | 61.8 | 19,200 | 6200 | 3.11 | 318 | 378 | 80 |
| 10 | Poly (PC/**1**/FDCA)[b] | TBT | 1 | 79.1 | 20,800 | 6200 | 3.34 | 306 | 388 | 72 |
| 11 | Poly (PC/**1**/FDCA)[c] | TBT | 3 | 79.1 | 27,500 | 11,700 | 2.35 | 294 | 395 | 74 |

General conditions: 3.35 mmol diol, 3.35 mmol DMFD/DMTA, 1 mol % catalyst, 190 °C N$_2$/1 h, 230 °C vacuum 1 mPa.
[a]Poly (PC/TPA) and poly (PC/FDCA) prepared in a vacuum for 3 h were subjected to purification by dissolving precipitation method.
[b]Copolymerization of PC-rich product mixture (PC, 82 and **1**13 %) obtained from catalytic funneling of lignin-derived PC/**1** with cellulose-derived DMFD. (Supplementary Note 3.5).
[c]Poly (PC/**1**/TPA) was prepared via copolymerization of lignin-derived PC/**1** with DMFD and DMTA (See Supplementary Note 3.4).
[d]Yield (%) = weight of collected product / weight of theoretical product × 100%.
[e]Molecular weight distribution was determined by GPC.
[f]T$_{5\%}$ and T$_{90\%}$ were determined by TGA characterization.
[g]T$_g$ was determined by DSC characterization.

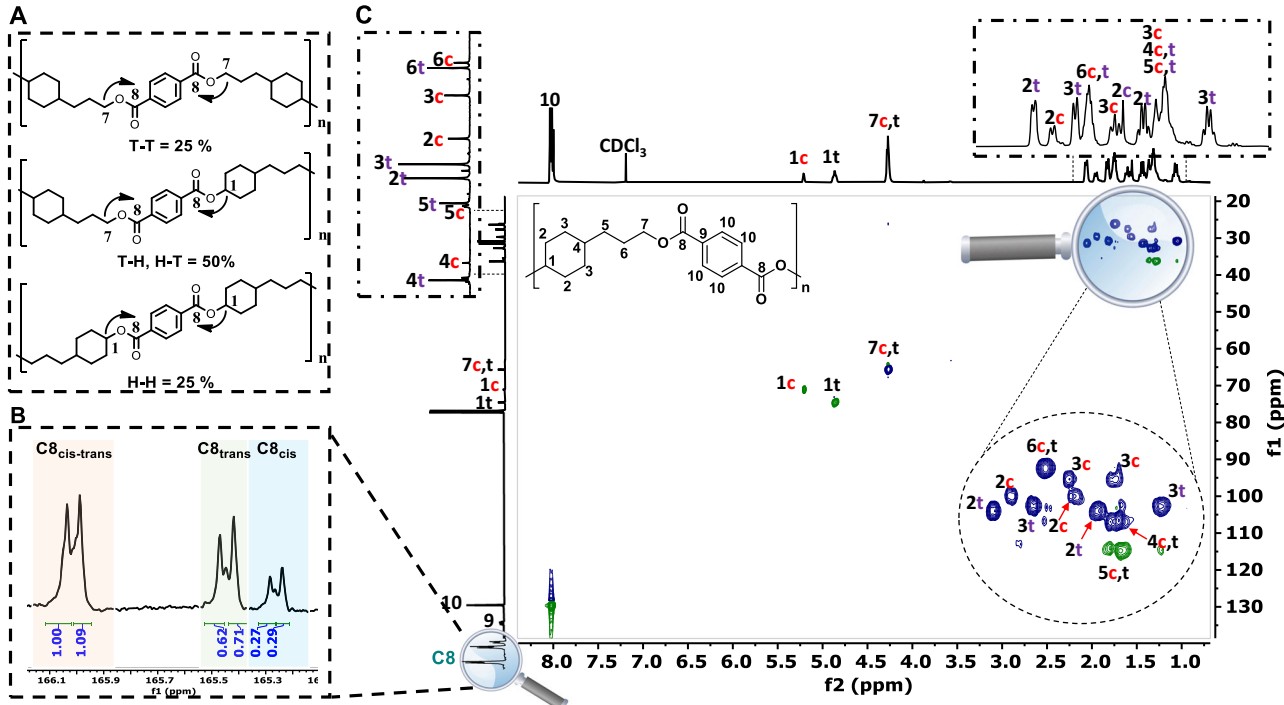

**Fig. 4 Structural characterization of poly (PC/TPA) based on NMR spectroscopy. A** Proposed unit connectivity of poly (PC/TPA) copolymerized using PC (1: 2 mixture of cis and trans isomers) and DMTA by forming three distinct dyads, namely head-to-tail (H-T), head-to-head (H-H), and tail-to-tail (T-T); **B** The identification and quantification of carbonyl carbon signals (C8) of poly (PC/TPA). Assignment of chemical shifts to the various connectivity units, namely cis, trans, and cis-trans; **C** 2D-HSQC characterization for the main-chain sequence of poly (PC/TPA).

or as a replacement. Gratifyingly, with glass transition temperatures ($T_g$) between 70–90 °C, both poly (PC/TPA) and poly (PC/FDCA) (Table 2, Entries 1–6) showed a thermal behavior comparable or better to that of commercial PET ($T_g = 67–80$ °C)[52]. Furthermore, these polymers also showed good thermal stability with decomposition temperature $T_{5\%} = 329$ °C and $T_{5\%} = 319$ °C under $N_2$, albeit lower than that of commercial PET (410 °C).

The physical and thermal properties of poly ($PC_{cis}$/TPA) (Table 2, Entry 7) and poly ($PC_{trans}$/TPA) (Table 2, Entry 8) were also compared with that of the synthesized poly (PC/TPA) (Table 2, Entry 1). While poly ($PC_{cis}$/TPA) was obtained in 84% yield, only 53% yield was reached for poly ($PC_{trans}$/TPA) indicating somewhat higher reactivity of the cis PC isomer in the polycondensation reaction. Interestingly, the molecular weight distribution of poly (PC/TPA) was higher than either of the polymers obtained from the pure isomers $PC_{cis}$ or $PC_{trans}$. A better heat resistance ($T_g$ value of 76 °C) was obtained for poly (PC/TPA) synthesized from a mixture of cis and trans PC isomers, while different $T_g$ values (72 °C versus 51 °C) were observed for the pure $PC_{cis}$ and $PC_{trans}$ analogues. The lower value of $T_g$ for poly ($PC_{trans}$/TPA), compared to poly ($PC_{trans}$/TPA) could be attributed to its lower Mw value (21.2 kg mol$^{-1}$ versus 16.8 kg mol$^{-1}$). The $T_{5\%}$ values were in the range of 310–320 °C for all poly (PC/TPA) variants. Gratifyingly, the obtained thermal characteristics demonstrate that there is no need for the separation of the mixture of cis and trans isomers of PC obtained by catalytic funneling prior to polymerization.

Next, polyesters from the aliphatic alcohol mixture of PC and **1** (PC, 82 and **1**, 13%) obtained from catalytic funneling of a standard equimolar mixture of 1 G and 1 S were prepared (Supplementary Note 3.4). The molecular weights of the corresponding poly (PC/**1**/TPA) (Mw = 19.4 kg mol$^{-1}$, Đ = 3.11) and poly (PC/**1**/FDCA) (Mw = 20.8 kg mol$^{-1}$, Đ = 3.34) were

somewhat lower than the polyesters prepared from pure PC (Table 2, Entries 9, 10 versus Entries 1, 4). This is reasonable, given that in the biomass-derived streams a mixture of PC (82% purity) and diol **1** is used for the copolymerization, whereby diol **1** bears an extra –OCH$_3$, which may influence reactivity. Therefore, future studies should elucidate the effect of the diol **1** comonomer on the polymer properties, and optimize the molecular weight by careful selection of reaction conditions. Also, further distillative purification of PC may be attempted on a larger scale.

**A comprehensive biorefinery strategy from beech wood to PET analogues and complementary products.** To show the scalability of our method in a comprehensive biorefinery context, we demonstrated access to PET analogues and complementary product streams such as light and heavy hydrocarbons as shown in Fig. 5 (see also Supplementary Note 3.5). Applying the previously developed two-step catalytic sequence (Cu20-PMO/methanol and Raney Ni/isopropanol) using 10 g beech lignocellulose, crude aliphatic bio-oil was obtained and was then subjected to careful fractional distillation (1 mpa, 100–120 °C) to deliver three distinct Fractions (A, B, and C), consisting of three specific product streams: 4-alkyl cyclohexanols (A), cyclohexane-diol derivatives PC and **1** (B), and higher boiling point dimers and oligomers (C).

Considering raw woody biomass as substrate, the number and difficulty of reaction and product isolation/purification steps will largely influence the overall feasibility of a biorefinery strategy. Here we demonstrate that the polymer building block PC can be obtained in a straightforward manner and high yield from raw biomass. In fact, Fraction B (Supplementary Fig. 77) obtained directly upon fractional distillation from EtOAc extracts, only consisted of diol PC and **1** in high purity (>99%), representing 15.3 wt% yield based on lignin content and could be directly subjected to copolymerization with methyl ester of FDCA to give

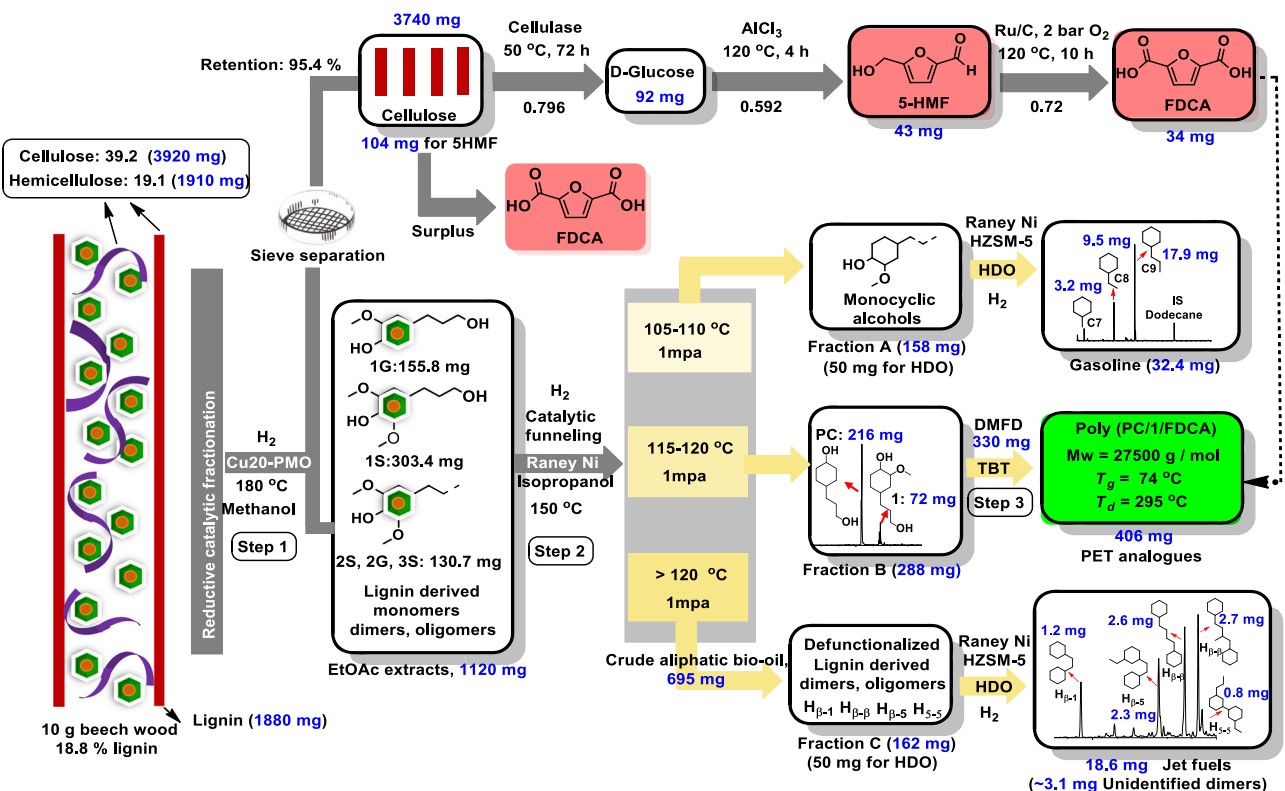

**Fig. 5 Comprehensive catalytic protocol for beech wood biorefinery for producing PET analogues and fuels.** Step 1: RCF of beech wood over Cu20-PMO catalyst (10 g beech wood, 2 g Cu20-PMO catalyst, 120 mL methanol, 40 bar $H_2$, 180 °C, 18 h) gives crude lignin oil and residual carbohydrates; Removal of impurities gives EtOAc extracts; Step 2: Catalytic funneling of EtOAc extracts (1120 mg EtOAc extracts, 2 g wet Raney Ni catalyst, 30 mL isopropanol, 150 °C, 12 h) delievers a mixture of crude aliphatic oil which is purified by distillation under 1 mpa at (100 –120 °C) to deliver three Fractions (A (100–105 °C), B (115 – 120 °C) and C (>120 °C)). PET analog synthesis: Reaction conditions (288 mg Fraction B, 330 mg DMFD (1.79 mmol), 1 mol% TBT catalyst, 190 °C $N_2$ for 1 h, 230 °C under vacuum for 3 h), Copolymerization of Fraction C with DMFD yields poly (PC/**1**/FDCA); Hydrodeoxygenation: Reaction conditions (50 mg Fraction A or C, 200 mg wet Raney Ni, 100 mg activated HZSM-5 co-catalyst, 20 mL cyclohexanol, 220 °C, 30 bar $H_2$, 4-6 h); HDO of Fraction A gives C7, C8, and C9 cyclic alkanes. HDO of Fraction C gives high-density cyclic alkanes. The hydrocarbons were quantified using the response of the flame-ionization detector (FID) and the response factors were estimated by the effective carbon number (ECN) method.

poly (PC/**1**/FDCA), that showed excellent and comparable molecular weight ($Mw = 27.5$ kg mol$^{-1}$, Đ = 2.35) (Table 2, Entry 11) and thermal properties ($T_g = 74$ °C and $T_{5\%} = 295$ °C), as poly (PC/FDCA) that was prepared using pure PC (for characterizations by $^1$H-NMR (Supplementary Fig. 78), GPC (Supplementary Fig. 79), TGA (Supplementary Fig. 80), and DSC (Supplementary Fig. 81).

Given that Fractions A and C consisted of more components, we aimed for extensive hydrodeoxygenation (HDO) to alkanes over Raney Ni and HZSM-5 co-catalyst at 220 °C, using cyclohexane as solvent. Thus, Fraction A was converted to gasoline range C7-C9 cyclic hydrocarbons, a mixture of 4-methyl-,4-ethyl-, and 4-propylcyclohexane (Supplementary Fig. 76). The heavier Fraction C was characterized by $^1$H-NMR (Supplementary Fig. 83), 2D-HSQC (Supplementary Fig. 85), and GC-FID (Supplementary Fig. 82), which confirmed the presence of oxygenated aliphatic dimers and oligomers. These were obtained from the extensive hydrogenation of the lignin-derived dimers and smaller oligomers present in the EtOAc extracts, originating from the β-1, β-5, β-β, and 5-5 linked aromatic dimer units present in hardwood[53,54]. Analysis of the products obtained upon HDO of Fraction C by $^1$H, DEPT NMR (Supplementary Figs. 83, 84), 2D-HSQC (Supplementary Fig. 86), and GC-FID (Supplementary Fig. 82), revealed successful removal of aliphatic –OH as well as –OMe groups to deliver higher Mw cyclic liquid hydrocarbons (only signals at 0.5–2.0 ppm observed). Further GC-MS analysis confirmed the presence of predominately

bicyclic hydrocarbons and a small amount of tricyclic and polycyclic hydrocarbons (Supplementary Fig. 82). These were quantified by the ECN method. Overall, from 50 mg Fraction C, 18.6 mg crude alkane product was obtained, consisting of predominately 1.2 mg $H_{β-1}$, 2.3 mg $H_{β-5}$, 5.3 mg $H_{β-β}$, 0.8 mg $H_{5-5}$, and c.a. 3.1 mg unidentified dimers or oligomers.

Overall, this strategy leads to an excellent valorization of beech lignocellulose by converting the lignin component to gasoline (C7–C9 hydrocarbons), high energy density alkane jet fuel (C14–C17 hydrocarbons), and PET analogue, accounting for an overall 29.5% carbon yield based on isolated yields (Supplementary Note 6).

Concerning the carbohydrate fractions, a sieve fractionation protocol to separate the catalyst from the cellulose residues was developed. Next, the catalyst-free solids (mainly containing cellulose), were subjected to a sequence of reaction steps to result in FDCA in a total mass yield of 32.7 wt%, part of which can be used to cover the FDCA necessary for the polymer synthesis (Supplementary Note 7). Alternatively, the surplus of cellulose could be also converted to bioethanol[55], or EG[22,23] while hemicellulose could be converted to valuable C5 platform chemicals such as furfural, or EG[24] (Supplementary Note 1).

*Techno-economic assessment.* On the basis of the experimental data, a preliminary techno-economic assessment (TEA) of the proposed biorefinery strategy shown in Fig. 5 (Supplementary Note 8) was performed, including the catalytic processing of

beech lignocellulose by RCF, followed by the fractionation of the obtained bio-oil and the catalytic processing of the respective fractions to final products.

Considering this early process stage, few basic assumptions with regard to fixed operating costs, utility costs and annual capital cost and solvent recovery, have been made in line with recent literature[39,56,57]. Gratifyingly, along with these mentioned assumptions and with the currently achieved product yields, the techno-economic evaluation shows a positive balance. More specifically, a 6.4% rate of return can be achieved at 99% solvent recovery. Returns are particularly sensitive to methanol and isopropanol recovery at 96 and 98% respectively. Overall, our analysis indicates that catalysts and solvent costs are the main drivers of operating cost, which is not surprising considering the lab-scale development stage of the process. FDCA and furfural form the most important revenue streams whilst fuels are neglectable in both volume and value. Hence, the profitability of the process is particularly dependent on future FDCA price assumptions. Since no mature FDCA market is yet existing, literature estimated revenue values were used[58–60].

Scale-up would focus on optimization of solvent demand and recovery as well as product yields. Here, especially the RCF step has the potential to be further optimized either in batch[61,62] or in continuous flow fashion[63,64] leading to both optimal solvent recoveries as well as near-theoretical 1 G/1 S yields that would lead to an estimated doubling of PC yield upon catalytic funneling. In addition, the yield of FDCA can also be further optimized to about 80% based on literature[65,66]. In fact, such an increase in yields would enable a profitable (7% return) operation of the process even under the assumption of the lowest possible FDCA prices discussed in the literature[58]. Another important aspect is to carefully assess the benefits of bio-based products compared to fossil-based ones, especially in relation to carbon neutrality and climate benefits. Current prices of fossil-based products PET or fuels we are aiming to substitute are still rather low and difficult to compete with, although this situation may gradually change in the future. However, assuming emission pricing in the range of 50 to 100 Euros per ton $CO_2$ released would add between 2 and 4% to the overall profitability.

*Circular economy approaches*. The recyclability of poly (PC/TPA) was evaluated by subjecting the material to alcoholysis using methanol, ethanol, 1-propanol, or *n*-butanol at 180 °C for 4 h, without any additives (Supplementary Note 4.1). The best results were obtained in methanol at 180 °C where 80% of PC and 85% of the methyl ester of TPA (DMTA) could be recovered. A lower yield (<40%) of PC and the corresponding ethyl and propyl terephthalates were obtained in ethanol and 1-propanol (Supplementary Fig. 88A) and no reactivity was found in n-butanol. Gratifyingly, methanolysis of poly (PC/TPA) at 190 °C resulted in 90 and 92% isolated yields of PC and DMTA obtained (Supplementary Fig. 88B).

With the optimized conditions in hand, this polymer was subjected to recycling (Supplementary Note 4.2). As shown in Fig. 6, poly (PC/TPA) was first synthesized and molded for possible applications. Then the produced material was subjected to methanolysis at 190 °C to give a crude mixture of PC and DMTA, characterized by GC-FID (Supplementary Fig. 90) and ¹H-NMR (Supplementary Fig. 91). Next, the crude mixture was repolymerized under the original conditions (1 mol% Zn(OAc)$_2$ catalyst, 190 °C N$_2$/1 h, 230 °C under vacuum (1 mPa/1 h) to regain poly (PC, TPA), which showed structural and thermal properties ($Mw = 16$ kg mol$^{-1}$, $T_g = 72$ °C, $T_{5\%} = 319$ °C) similar to the virgin polymer (Supplementary Figs. 92–94). These experiments demonstrate the feasibility of our PET analogs for the circular economy.

## Discussion

Reductive catalytic fractionation (RCF) is able to convert raw lignocellulosic biomass into phenolic monomers, aromatic C–C bonded dimers, oligomers, and high quality (hemi)cellulose streams. This work demonstrates an RCF-based biorefinery approach to produce PET analogues made entirely of wood, together with gasoline and jet-range fuels and fuel additives. Central to the strategy is the catalytic funneling of aromatic monophenols 1 G and 1 S into 4-(3-hydroxypropyl) cyclohexan-1-ol (PC) by highly selective Ni-catalyzed hydrogenation/ defunctionalization. When applying this method to the complex bio-oil obtained upon RCF from native lignin, a mixture of PC and other cyclic and bicyclic aliphatic alcohols are obtained in the same pot, which are separated by fractional distillation into three distinct cuts based on their boiling point. Thus, PC along with diol 1 (PC, 78% and 1, 22%) are isolated in high purity and copolymerized with cellulose-derived methyl esters of FDCA and TPA. The lighter and heavier fractions can be further processed by HDO to petrol and jet-fuel range alkanes, respectively. Overall, an estimated lignin utilization efficiency of 29.5% can be reached based on isolated product yields, which may be further improved by adjusting the RCF step at the front end of the process. An interesting research direction would be to attempt the use of Cu20-PMO or other suitable catalysts in a continuous flow operation[63,64], thereby focusing on maximizing yield and selectivity of 1 G and 1 S to afford near-theoretical yield of PC and optimal solvent consumption and recovery. In fact, the Ni-catalyzed funneling strategy developed here can be used for any bio-oil obtained from RCF[58] or other prominent cutting-edge depolymerization efforts[59].

In addition, FDCA can be obtained directly from the catalyst-free cellulose-rich residues in up to 33% yield (cellulose basis), matching or surpassing the needed amount for the copolymerization. Besides, any surplus of carbohydrates could be converted to ethylene glycol, or other aliphatic diols, which may be used as a comonomer to improve the mechanical properties of the obtained polymer.

The overall lignocellulose processing approach presented here, consisting of industrially feasible catalytic steps and purification methods, enables the smooth processing of complex raw biomass feed into PC diol which can be readily incorporated into the fully bio-based polyesters poly (PC/1/TPA) and poly (PC/1/FDCA). These polymers display molecular weight range and thermal properties comparable to or better than conventional PET and can be recycled into the virgin monomers. Preliminary studies indicate that further optimization of the mechanical properties is necessary, for example by incorporating of other bio-based diacids or diols or other suitable plasticizers. Also, the optimization of the molecular weights and polydispersity should be carried out with further purified PC diol. Nonetheless, the already measured competitive physical and chemical properties of the polymers, together with the favorable techno-economic assessment, in addition to the straightforward method of producing PC that avoids any extensive purification effort or column chromatography, makes this process interesting for considering future upscaling and the proposed PET and PEF analogues derived entirely from non-edible sources, may become promising candidates for the circular economy.

## Methods
**Materials and reagents**. Pine lignocellulose was purchased from Bemap Houtmeel B.V. Poplar and beech lignocellulose were obtained from a local woodshop (Dikhout, Groningen, the Netherlands). Chemicals were used as received unless otherwise specified. Raney Ni 4200, Palladium on carbon, Nickel on carbon, Ruthenium on carbon, 65% Ni/SiO$_2$-Al$_2$O$_3$, Titanium (IV) butoxide (TBT), Zn(OAc)$_2$ were purchased from Sigma-Aldrich, 10% Ni/γ-Al$_2$O$_3$ was purchased from Riogen, USA.

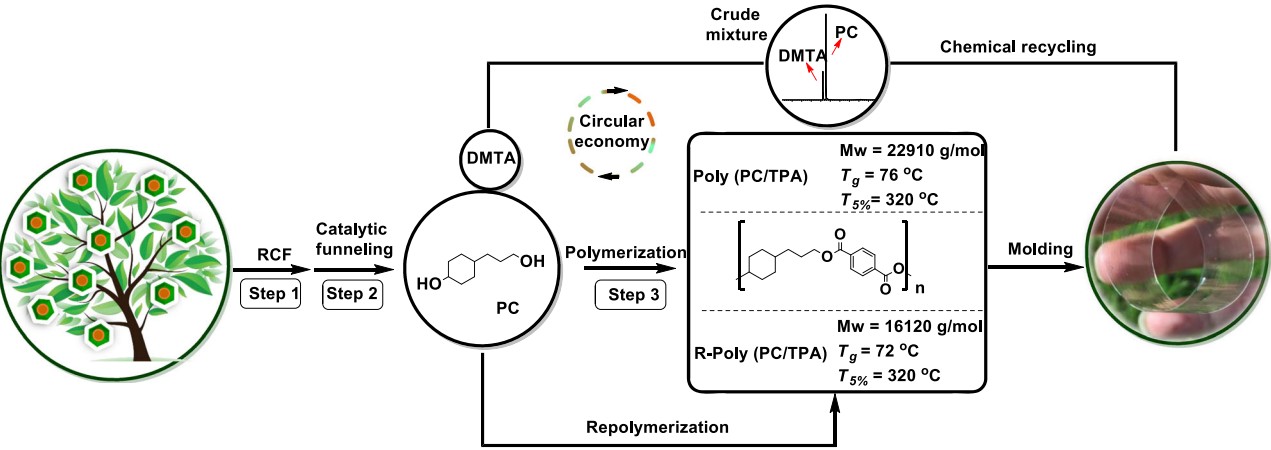

**Fig. 6 A circular economy strategy for plastic recycling.** Step 1: RCF of lignocellulose over previously developed Cu20-PMO catalyst; Step 2: Catalytic funneling of RCF crude mixture over Raney Ni catalyst using isopropanol as solvent.; Step 3: The copolymerization of PC with DMTA to give poly (PC/TPA) using Zn(OAc)₂ catalyst; One-pot methanolysis of poly (PC/TPA) to recycle monomers PC and DMTA; The crude mixture from depolymerized poly (PC/TPA) was repolymerized using Zn(OAc)₂ catalyst to give -poly (PC/TPA).

**Preparation of Cu20-PMO catalyst**. Synthesis of Cu20-PMO catalyst was prepared according to our previously reported procedure[1]. In a typical procedure, a solution containing AlCl₃·6H₂O (12.07 g, 0.05 mol), Cu(NO₃)₂·2.5H₂O (6.98 g, 0.03 mol), and MgCl₂·6H₂O (24.4 g, 0.12 mol) in deionized water (200 mL) was dropwise added to a solution containing Na₂CO₃ (5.30 g, 0.05 mol) in water (300 mL) at 60 °C under vigorous stirring. The pH value was always kept between 9 and 10 by the addition of small portions of a 1 M solution of NaOH. The mixture was vigorously stirred at 60 °C for 72 h. After cooling to RT, the light blue solid was filtered and resuspended in a 2 M solution of Na₂CO₃ (300 mL) and stirred overnight at 40 °C. The catalyst precursor was filtered and washed with deionized water until chloride free. After drying the solid for 6 h at 100 °C followed by the calcination at 460 °C for 24 h in air, 9.5 g of Cu20-PMO was obtained.

*Reductive catalytic fractionation of lignocellulose.* Reductive catalytic fractionation of lignocellulosic biomass was carried out in a 100 mL high-pressure Parr autoclave equipped with an overhead stirrer. Typically, the autoclave was charged with 0.4 g of Cu20-PMO catalyst, 2 g of lignocellulose (beech, pine, or poplar), and methanol (20 mL) as a solvent. The reactor was sealed and pressurized with H₂ (40 bar) at RT. The reactor was heated to 180 °C and stirred at 400 rpm for 18 h. After completion of the reaction, the reactor was cooled to RT. Then 0.1 mL solution was collected through a syringe and injected into GC-MS or GC-FID after filtration through a PTFE filter (0.45 μm). The solid was separated from the solution by centrifugation and subsequent decantation and additionally washed with methanol (3 × 20 mL). The methanol extracts were combined in a round bottom flask and the solvent was removed in vacuo. The crude aromatic bio-oil was dried in a desiccator in vacuo overnight and was further used as specified below.

*Fractionation of crude aromatic bio-oil.* To the obtained crude aromatic bio-oil EtOAc (20 mL) was added and it was stirred overnight at RT, which resulted in precipitation of a brownish-colored solid. The suspension was then transferred into a 20 mL centrifuge tube. The solid was separated by centrifugation and decantation and additionally washed with EtOAc (2 × 20 mL). The EtOAc soluble fractions were combined in a round bottom flask and the solvent was removed in vacuo. The soluble and insoluble fractions were additionally dried in vacuo under constant weight. The soluble fraction was washed with a small amount of saturated NaHCO₃ (1 × 10 mL) and brine (2 × 10 mL) and dried over anhydrous MgSO₄ and further used as specified below.

*Catalytic demethoxylation and hydrogenation of model compound 1G.* Demethoxylation/hydrogenation of model compound 1 G to PC was carried out in a 100 mL high-pressure Parr autoclave equipped with an overhead stirrer. Typically, the autoclave was charged with 1 g wet Raney Ni catalyst, 0.2 g (1.1 mmol) 1 G, 15 mL isopropanol, and 20 mg dodecane as internal standard. The reactor was sealed and pressurized with H₂ (10 bar) at RT. The reactor was heated to 120 °C and stirred at 400 rpm for 2 h. After completion of the reaction, the reactor was cooled to RT. Then 0.1 mL solution was collected through a syringe and injected into GC-MS or GC-FID after filtration through a PTFE filter (0.45 μm). The Raney Ni was separated from the solution by centrifugation and subsequent decantation and additionally washed with isopropanol (3 × 20 mL). Then the isopropanol soluble fractions were combined in a round bottom flask and the solvent was removed in vacuo. The crude product was dried in a desiccator in vacuo overnight and was further used as specified below.

*Synthesis of PET analogues derived from PC and comonomer DMFD or DMTA.* Copolymerization of model compound PC with DMFD and DMTA was performed using an equal molar ratio of PC and DMTA or DMFD over Zn(OAc)₂ or Titanium (IV) butoxide (TBT) catalyst. For example, a 100 mL three-neck flask equipped with a magnetic stirrer and reflux condenser was charged with 0.53 g (3.35 mmol) of PC diol, 0.66 g (3.35 mmol) of DMTA, and 1 mol% (0.0057 g) Zn(OAc)₂ catalyst. The esterification reaction was performed at 190 °C for 1 h under nitrogen flow. Then, the reaction temperature was increased to 230 °C, the pressure was reduced to 1 mPa using an oil pulp for 1 and 3 h, respectively. After that, the reaction mixture was cooled down to RT and the pressure was returned to atmospheric pressure by introducing nitrogen. The obtained solid was characterized by NMR, DSC, TGA, FT-IR, and GPC.

*Methanolysis of poly (PC/TPA).* Chemical recycling of the synthesized poly (PC/TPA) was carried out in a 100 mL high-pressure Parr autoclave equipped with an overhead stirrer. Typically, the autoclave was charged with poly (PC/TPA) (0.2 g), dodecane (20 mg), and methanol (30 mL). The reactor was sealed and flushed with N₂ three times. The reactor was heated to 190 °C and stirred at 400 rpm for 4 h. After completion of the reaction, the reactor was cooled to RT. The PC-rich mixture was then isolated and purified by column chromatography using EtOAc/n-hexane (1:2 to 1:1 to 2:1).

### Comprehensive biorefinery strategy for the conversion of beech wood to PET analogs and complementary products

*Step 1.* A large-scale reductive catalytic fractionation (RCF) setup using beech wood was carried out over Cu20-PMO catalyst under the following reaction conditions: 10 g beech wood, 2 g catalyst, 180 °C, 120 mL methanol, 40 bar H₂, 18 h. RCF of beech wood gave crude depolymerized lignin oil (1531 mg). To the crude depolymerized lignin oil 100 mL of EtOAc was added and the suspension was stirred overnight. Lignin residues and small amounts of sugars were precipitated as brownish-colored solid and separated by centrifugation, decantation and washed with EtOAc (20 mL). The EtOAc soluble fraction was washed immediately with small amount of saturated NaHCO₃ (1 × 10 mL) and brine (2 × 20 mL) to deliver EtOAc extracts. Then the EtOAc extracts were transferred in a round bottom flask and the solvent was removed in vacuo. The EtOAc extracts was dried in a desiccator in vacuo overnight and was further used as specified below.

*Step 2.* The catalytic funneling of EtOAc extracts were carried out in a 100 mL high-pressure Parr autoclave with an overhead stirrer. Typically, the autoclave was charged with 2 g Raney Ni catalyst, 1120 mg of EtOAc washings, and 20 mL isopropanol. The reactor was sealed and pressurized with H₂ (30 bar). The reactor was heated and stirred at 150 °C for 10 h. After the reaction, the reactor was cooled to room temperature and the solvent was removed in vacuo to deliver an aliphatic alcohol mixture. The aliphatic alcohol mixture was subjected to distillation at a temperature range between 100–120 °C using the Kugelrohr apparatus under vacuum 1 mPa, to provide Fraction A, Fraction B, and Fraction C.

*Step 3.* To the Fraction, B was added 322 mg (1.75 mmol) of DMFD and 1 mol% TBT catalyst in a 100 mL three-neck round bottom flask. Then the crude mixture was heated to 190 °C for 1 h under nitrogen flow. The pressure was slightly reduced to 1 mPa using an oil pulp and the mixture was heated to 230 °C. The reaction was

considered to be complete under vacuum for 3 h. Finally, 406 mg (81.5% yield) of poly (PC/1/FDCA) was obtained. The Fraction A were selectively hydro-deoxygenated (200 mg wet Raney Ni, 100 mg HZSM-5, 20 mL cyclohexane, 220 °C, 30 bar $H_2$, 4 h), to give predominantly 4-ethyl cyclohexane (C8) and 4-propyl cyclohexane (C9), as well as a small amount of 4-methyl cyclohexane (C7). The fraction C was subjected to hydrodeoxygenation (HDO), over Raney Ni and HZSM-5 co-catalyst (200 mg wet Raney Ni, 100 mg HZSM-5, 20 mL cyclohexane, 220 °C, 30 bar $H_2$, 6 h) to give predominantly dicyclohexane derivatives. The amount of the respective hydrocarbons was determined by the ECN method where the response factor was assumed as 1.0.

## Data availability

The authors declare that all of the data that support the findings of this study are available within the article and its Supplementary Information files or from the corresponding author upon reasonable request.

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

## Acknowledgements

K.B. is grateful for financial support from the European Research Council, ERC Starting Grant 2015 (CatASus) 638076, and ERC Proof of Concept Grant 2019 (PURE) 875649.

This work is part of the research program Talent Scheme (Vidi) with project number 723.015.005, which is partly financed by The Netherlands Organization for Scientific Research (NWO). X.W. is grateful for financial support from the China Scholarship Council (grant number 201808330391). T.S. thanks COST Action FUR4Sustain- European network of FURan-based chemicals and materials FOR a Sustainable development, CA18220, supported by COST (European Cooperation in Science and Technology).

## Author contributions

X.W. designed and performed all related experiments. X.W. also contributed to data collection, data analysis, structural analysis of polymers, and manuscript preparation. Z.S. conceived the idea of targeting **PC** by funneling for polymer purposes, performed carbohydrate conversion, polymer mechanical property analysis, and lignocellulose composition analysis. M.V.G. performed reaction kinetics analysis, structural analysis of polymers, and manuscript revision. T.S. performed TEA analysis and contributed to manuscript revisions. K.B. conceived the research, designed experiments, contributed to figure design, supervised the research, and wrote the manuscript. All authors commented on and approved the final manuscript.

## Competing interests

The authors declare no competing interests.
