## [Peer Review File · Nature Communications]

Fully lignocellulose-based PET analogues for the circular economyREVIEWER COMMENTS

Reviewer #1 (Remarks to the Author):

“Fully lignocellulose-based PET analogues for the circular economy” by Wu, Galvin, Sun, and Barta is a complete process approach to addressing the sourcing, synthesis, and recyclability of a fully bio-based PET replacement. This paper is important for bridging the gap between the fundamental problems with waste accumulation and petroleum-derived product dependence and implementing bio-based alternatives into the market with a bioeconomy at the forefront. Starting from whole biomass, the authors performed reductive catalytic fractionation (RCF) on various sources of biomass using a copper based porous metal oxide catalyst to convert lignin to 4-propanolsyringol and 4-propanolguaiacol aromatic monomers with high selectivity. These monomers were reacted with a Raney nickel catalyst to produce the monomer PC and the small molecule was polymerized with TPA or FDCA to yield PET analogous materials with similar thermal properties. Next, these materials were recycled to methylated monomers through alcoholysis and repolymerized to demonstrate the retention of thermal properties and the entire biorefinery process was highlighted at the conclusion of the work.

The comprehensive approach in this work is appreciated and thoroughly demonstrated. This is a challenging topic, and though the chemistry for each reaction step exhibited is established, the complete display of starting from whole biomass through the recycling of a PET analogue gives this work merit. There are a few limitations to the work that potentially (slightly) reduce the potential impact to the field. These limitations are: 1) the authors did not utilize the isolated carbohydrate fraction from RCF to produce the TPA or FDCA needed to produce their fully bio-based PET analogues (this is a very minor point), 2) they did not perform mechanical property testing on their synthesized polymers to confirm their industrial relevance (this is a fairly major point), and 3) the thermal properties do not outperform PET in a meaningful way (this is a minor point).

General comments to address in the manuscript are:

The resolution of the figures in the main manuscript appears to have not been properly formatted – this might be a PDF conversion issue (I am assuming it is?). For example, Fig 1 b, there is an overlay of a picture with what seem to be previous images. Probably this is an easy fix, but wanted to note it in case note.

This issue of formatting is also prevalent in the supplementary information. Both the text and images seem to have error in their placement/resolution.

Also, the amount of detail provided in the methods section of the main manuscript should be improved. Additional information should be included on the overall reactor/reactions procedures used, especially for the scaled up reactions shown in Fig 5 (10g). The details on the fractionation methods for the RCF crude mixtures provided in the SI should be included in the main manuscript as frequent references to “crude 1” and “fraction 3” are difficult to keep track of.

There are some more refined points, I suggest that the authors address:

- In the authors' previous paper (ref 43 in main paper), their proposed strategy to liberate the used catalyst from carbohydrate rich pulp is to react the pulp/catalyst at 300C in supercritical methanol, which they demonstrated would convert the pulp to a wide distribution of aliphatic alcohols. So this strategy cannot be utilized for the proposed strategy in this work to utilize the carbohydrate pulp after reaction to produce the necessary TPA or FDCA to produce the PET analogue polymers. I think some discussion on how to separate the catalyst from the carbohydrate pulp by the authors is needed especially with the emphasis of the work being the full utilization of biomass.

o If the authors thought is that this process could be done in dual-bed flow system, similar to those demonstrated by Anderson et al. Joule, 2017, <https://doi.org/10.1016/j.joule.2017.10.004> &

Kumaniaev et al. Green Chemistry, 2017, DOI: 10.1039/c7gc02731a to avoid the need to liberate the catalyst from pulp, would the conditions used in this paper (180°C, 18 h) still hold to produce RCF monomers in the shorter residence times typically utilized in these flow systems?

o Have the authors considered/tested other catalysts for the RCF step? Other catalysts like Pd/C (Van den Bosch et al. Energy Environ. Sci., 2015, DOI: 10.1039/c5ee00204d) have been known to produce RCF oils with high selectivity to 4-propanol syringol/guaiacol (~90%) as well but at reaction times similar to those used in those flow systems (~3hr).

- Figure 2C: It seems the overall selectivity to PC is about the same when running in isopropanol, THF, Me-BuOH, and Me-THF, and conversion limits the yield to PC with these other solvents. However, it seems the product distribution is slightly more favorable when using THF since 1H is one of the only side products and this should be more advantageous to forming additional amounts of PC than products 2 and 3 seen in the reactions with isopropanol. Did the authors run the reaction in THF for longer reaction times to reach full conversion? Would reaching 100% conversion in THF give similar PC yields to reactions in IPA and still have 1H present?

- Table 1/Figure 3: The authors calculated selectivity in terms of moles of propanol syringol/guaiacol converted to PC. With the amount of each compound written in terms of mass in Figure 3 and with the conversion being in terms of mass for the rest of the manuscript, this distinction is not very intuitive to determine right away. The authors should include in the table caption for Table 1 that selectivity is defined on a molar basis or cite the equation used in supplement section 1.3.

- Figure 5: The authors note they achieve an overall mass efficiency of 36% from lignin to PC, gasoline and jet fuel chemicals. I assume the majority of the mass loss is from removal of oxygen from HDO reactions. I think it would be helpful to rephrase/calculate what the efficiency is on a per carbon basis after RCF, catalytic funneling, and further upgrading reactions.

- The polymerizations were carried out under typical conditions, though the industry and other relevant reports (ref 9, 28, 29 of the main manuscript) of lignin-derived polyesters use Sb₂O₃ as the catalyst and some explanation as to the departure from that compound would be appreciated, or the testing of that catalyst.

- Molecular weights are comparable along with thermal properties of PET; however, it is notable that these molecular weights are obtained with model compounds in Table 2, Entries 1-8. Entries 9 and 10 with monomer PC isolated from biomass have lower molecular weights and higher dispersities. This is possibly due to the monomer purity being 95%. For polycondensations to achieve higher molecular weight, purity of >99% is typical. It would be gratifying to see if improved monomer purity achieved higher molecular weights with lower dispersity as well as molecular weights after polymerization of PC (produced from RCF) at 3 hours.

- Mechanical and barrier properties associated with PET are extremely relevant for industrial applications. For instance, PEF has not become an attractive alternative due to brittleness and lack of strain-hardening, which limits the application of this material in a pressurized environment like a soda/water bottle. Mechanical property testing and barrier property measurements would be highly impactful for claiming industrial relevance. This is likely an experiment that should be done for this paper.

Reviewer #2 (Remarks to the Author):

The new strategy producing fully lignocellulose-based PET alternatives with excellent properties and recyclability seems promising. Extensive characterization of products, reaction kinetics and reaction mechanisms have been performed. The experimental results on the production of DMFT and DMTA from the carbohydrate pulp part seem a gap in this work. Process economics is another missing piece. These are the major comments and few more minor comments to be addressed before

considering it for publication.

1. Improve the experimental results on the production of DMFT and DMTA from the carbohydrate pulp.
2. Add process economics.
3. Step 3 is not shown in the Figure 1.
4. In Figure 5 and Fig S74, hemi(cellulose) content should be 8120 mg instead of 9120 mg.
5. Inconsistency of unit, "bar" vs. "MPa".
6. Do the authors look into what makes Raney Ni more superior than the other heterogeneous catalysts for chemical funneling.
7. Figure 2 labels and axes labels are hard to read.
8. Optimized temperature was said to be 140oC. However, Figure 2D only shows results up to 120oC
9. Typo in conclusion, "PTA" should be "TPA".

Reviewer #3 (Remarks to the Author):

Some grammatical issues to be addressed include:

Page 3: "using Raney nickel led to" (not "using Raney nickel lead to")

Page 3: "As a comparison," (add "a")

Figure 1a: The PEVA structure drawn is based on syringic acid and probably should be PESA. The Tg of 82 also suggests this is the syringic acid polymer. PEVA probably applies to the vanillic acid variant, but it has a lower Tg. There is something under the wood graphic that probably was meant to be in front of the wood graphic.

Page 9: "in applications similar to those of PET or as" (not "in applications similar to the application are of PET or as")

Page 9: "glass transition temperatures (Tg) between" (add "s")

The PC yield is given as 85% from 1G/1S in Figure 1. But the text claims "56.4% efficiency," which I assume is the yield from lignin. But the yields of PC in Figure 3 are 15% maximum. Yet another "estimated" value of 36.2% "utilization efficiency" is mentioned in the conclusions. The overall yield from the lignin starting material should be better described in the introduction—perhaps the mass yield from the biomass wood starting material. A 15% production of one molecule from lignin seems to be a great result—much higher than the state of the art alternative procurements, such as vanillin from spruce (perhaps 1% from wood itself). But has the PC been separated and purified? What is the real percent yield of pure, isolated PC?

Figure 1b: And the left "OH" group of PC should be "HO" as drawn.

Figure 1b: Molecular weights are provided with too many significant figures. Typically the tens and one positions are both zero, with most instruments. Also, Tg values usually have no decimal places. Conventionally, one would describe a copolymer as poly(PC/TPA) and not Poly (PC, TPA).

Page 8: "dyad" (not "diad", according to the dictionary and experienced polymer chemists)

Page 9: The Tg of 51°C for the trans polymer seems rather low compared to the other stereochemical variants. It is also has the lowest molecular weight among the PC/FDA and PC/DMT variants.

Perhaps the Mw is too low to provide a realistic Tg value here and a comment could be added to warn this.

Page 9: Poly(PC/1/FDCA) is lauded as being highly biobased and having a high Tg value near 74 °C. FDCA, while biobased, is still very expensive to produce. Moreover, poly(ethylene glycol/FDCA) (polyethylene furanoate, PEF), the long-promised commercial product from FDCA has an even higher Tg near 86 °C—and uses a currently available and inexpensive bio-based diol, ethylene glycol. So while there are certainly some commercial possibilities with PC, it performs no better than ethylene glycol. One advantage that could be mentioned is that EG derives from a food source, Brazilian sugar, but PC derives from a non-food source, lignin.

Overall the authors have presented a novel biorefinery system that converts wood to fuels and useful packaging plastics. The technique of reductive catalytic fractionation (RCF) was adapted very well to

the deconstruction of woody biomass. There is considerable novelty and value in the (relatively) high conversion of lignin to the aliphatic diol, PC (hydroxypropylcyclohexanol). The creation PET mimics with PC was clearly demonstrated as such polymers match or slightly excel the T_g of PET. In turn, depolymerization (via methanolysis) of the PET mimics was demonstrated and thereby, shown to be a viable step in the circular recycling of these materials. I recommend publication in Nature Communications after attention is paid to the points made above.

Reviewer #1 (Remarks to the Author):

“Fully lignocellulose-based PET analogues for the circular economy” by Wu, Galvin, Sun, and Barta is a complete process approach to addressing the sourcing, synthesis, and recyclability of a fully bio-based PET replacement. This paper is important for bridging the gap between the fundamental problems with waste accumulation and petroleum-derived product dependence and implementing bio-based alternatives into the market with a bioeconomy at the forefront. Starting from whole biomass, the authors performed reductive catalytic fractionation (RCF) on various sources of biomass using a copper based porous metal oxide catalyst to convert lignin to 4-propanolsyringol and 4-propanolguaiacol aromatic monomers with high selectivity. These monomers were reacted with a Raney nickel catalyst to produce the monomer PC and the small molecule was polymerized with TPA or FDCA to yield PET analogous materials with similar thermal properties. Next, these materials were recycled to methylated monomers through alcoholysis and repolymerized to demonstrate the retention of thermal properties and the entire biorefinery process was highlighted at the conclusion of the work.

The comprehensive approach in this work is appreciated and thoroughly demonstrated. This is a challenging topic, and though the chemistry for each reaction step exhibited is established, the complete display of starting from whole biomass through the recycling of a PET analogue gives this work merit. There are a few limitations to the work that potentially (slightly) reduce the potential impact to the field. These limitations are: 1) the authors did not utilize the isolated carbohydrate fraction from RCF to produce the TPA or FDCA needed to produce their fully bio-based PET analogues (this is a very minor point), 2) they did not perform mechanical property testing on their synthesized polymers to confirm their industrial relevance (this is a fairly major point), and 3) the thermal properties do not outperform PET in a meaningful way (this is a minor point).

We appreciate the positive evaluation of our work. Indeed, our key goal here was to demonstrate an integrated biorefinery scheme that displays high practicality, but also provides new scientific insight. We believe, that many more approaches and new routes toward PET replacements are still needed, and here we provide one possible alternative. In our view, the main advantage of our method is that it is able to incorporate the lignin as well as the cellulose constituent into the bio-based polyester, and is industrially relevant, and efficient. We appreciate the valuable suggestions and comments of the expert **Reviewer #1** – we have addressed these in detail below.

General comments to address in the manuscript are:

Question #1, Reviewer #1: The resolution of the figures in the main manuscript appears to have not been properly formatted – this might be a PDF conversion issue (I am assuming it is?). For example, Fig 1 b, there is an overlay of a picture with what seem to be previous images. Probably this is an easy fix, but wanted to note it in case note.

Answer #1, Reviewer #1: Indeed, this was a formatting mistake as result from a PDF conversion issue; we have now added a new high-resolution **Figure 1b** in the revised manuscript.

Question #2, Reviewer #1: This issue of formatting is also prevalent in the supplementary information. Both the text and images seem to have error in their placement/resolution.

Answer #2, Reviewer #1: We apologize for the formatting issues in the supplementary information that we have overlooked despite many rounds of checking the SI document. All formatting issues have been addressed in the revised supplementary information.

Question #3, Reviewer #1: Also, the amount of detail provided in the methods section of the main manuscript should be improved. Additional information should be included on the overall reactor/reactions procedures used, especially for the scaled up reactions shown in Fig 5 (10g). The details on the fractionation methods for the RCF crude mixtures provided in the SI should be included in the main manuscript as frequent references to “crude 1” and “fraction 3” are difficult to keep track of.

Answer #3, Reviewer #1: We highly appreciate the reviewer’s valuable comment. These details were in the supporting information mainly to adhere to the word count limitations. But in order to make this clearer, we have added an additional sub-heading into the experimental section in the main manuscript, regarding the comprehensive biorefinery strategy for the conversion of beech wood to PET analogues and complementary products, which is shown on Figure 5 (10 g run). In addition, the frequent references to the crude 1, crude 2 and fraction 3 have been remade and renamed in the revised manuscript. For example, crude 1, fraction 3, were replaced by crude aromatic bio-oil and EtOAC extracts, respectively.

There are some more refined points, I suggest that the authors address:

Question #4, Reviewer #1: In the authors’ previous paper (ref 43 in main paper), their proposed strategy to liberate the used catalyst from carbohydrate rich pulp is to react the pulp/catalyst at 300C in supercritical methanol, which they demonstrated would convert the pulp to a wide distribution of aliphatic alcohols. So, this strategy cannot be utilized for the proposed strategy in this work to utilize the carbohydrate pulp after reaction to produce the necessary TPA or FDCA to produce the PET analogue polymers. I think some discussion on how to separate the catalyst from the carbohydrate pulp by the authors is needed especially with the emphasis of the work being the full utilization of biomass.

Answer #4, Reviewer #1: We appreciate the comments on our previous work (Nat. Catal. 2018, 1, 82-92) where we liberated the catalyst by further converting the reaction solids to aliphatic alcohols in supercritical methanol. As Reviewer 1 points out, the latter is not the preferred strategy when **TPA** and **FDCA** are the desired targets from cellulose. In fact, our here developed two-step method, consisting of **RCF**, followed by the newly established catalytic funneling toward **PC** diol, may be conducted with a number of catalyst choices in the first, RCF step, Cu₂₀PMO being one of them. We have mainly used this catalyst as it is a working standard in our laboratory. Regarding the RCF step, catalyst separation and re-use has been established in the field, for example with the use of a catalyst cage (Green Chem. 2017, 19, 3313–3326) or separation with a magnet in the case of Raney Nickel catalyst (Angew. Chem. Int. Ed. 2014, 53, 8634-8639).

To fully satisfy the Reviewer’s point, here we have developed a straightforward protocol using beech sawdust with a size of more than 1 mm, taking advantage of sieve fractionation, inspired by (Song et al. Biotechnol. Biofuels 2020, 13, 2). Here, after **RCF** the spent Cu-PMO catalyst was separated from the carbohydrate pulp first through mesh screening. Then, the remaining cellulose rich solids were subjected to further ultrasonic treatment in water to get rid of the catalyst residues. After applying this method, the isolated reaction solids (mainly cellulose) were subjected to ICP analysis, which showed minimal Cu contamination (1.45 mg Cu/ g carbohydrate) that confirmed the removal of Cu-PMO catalyst. The carbohydrate containing solids were subjected to further chemical treatment to yield **FDCA** in a mass yield of 32.7 wt. % on a cellulose basis.

We have added **Supplementary Note 1** that describes the separation and conversion of isolated carbohydrates into **FDCA**. In the main text, clarifying comments were added and **Figure 5** was modified accordingly.

Question #5, Reviewer #1: If the authors thought is that this process could be done in dual-bed flow system, similar to those demonstrated by Anderson et al. *Joule*, 2017, & Kumaniaev et al. *Green Chemistry*, 2017, **19**, 5767-5771 to avoid the need to liberate the catalyst from pulp, would the conditions used in this paper (180°C, 18 h) still hold to produce **RCF** monomers in the shorter residence times typically utilized in these flow systems?

Answer #5, Reviewer #1: We thank Reviewer #1 for the insightful comment. Application of a flow system for Cu-PMO mediated RCF sounds very interesting, that could be implemented in our future studies. At the moment we do not have the possibility to build such a dual bed flow system and we did not consider it for this work. In our future studies, we would test a range of different catalysts and make sure to overcome common disadvantages related to the product to solvent ratio. Also, related to the rather fine ('fluffy') nature of the Cu-PMO catalyst, we would investigate catalyst morphology details either by implementing specific physical processing of the catalyst into larger pellets, or by e.g. SBA-15 based templated synthesis.

To address this point, a comment was inserted about flow systems as an exciting future direction into the conclusion section of the manuscript.

Question #6, Reviewer #1: Have the authors considered/tested other catalysts for the RCF step? Other catalysts like Pd/C (Van den Bosch et al. *Energy Environ. Sci.*, 2015, DOI: 10.1039/c5ee00204d) have been known to produce RCF oils with high selectivity to 4-propanol syringol/guaiacol (~90%) as well but at reaction times similar to those used in those flow systems (~3hr).

Answer #6, Reviewer #1: In this study we have focused on the non-noble metal based and inexpensive Cu-PMO system to perform RCF, of which we already have detailed knowledge.

However, as earlier mentioned, our catalytic funneling strategy developed here is able to accommodate versatile RCF product streams, meaning that different catalysts can be used for the RCF step. Most ideal choices would be RCF methods that provide a mixture of **1G/1S** in high enough yield and selectivity, and as Reviewer #1 points out, this in combination with flow systems would be an excellent choice.

To address this further point, we have tested commercially available Pd/C for the RCF of beech wood and found a high selectivity (up to 94 %) and higher yield to **1G/1S**, compared to the Cu₂₀PMO system. The results are summarized in **Table S10**.

It is true that the RCF step should be optimized, as the resulting **1G/1S** yield strongly influences the yield of the desired **PC** diol, which has an influence on the overall techno economics of the system. The performed TEA analysis suggests that the overall economic feasibility is sensitive to the **PC** yield, while the cost of the catalyst is not detrimental as long as it can be sufficiently recycled. A comment regarding the use of different catalyst systems for the RCF step and the importance of maximizing the **1G/1S** yield have been added to the manuscript.

Question #7, Reviewer #1: Figure 2C, It seems the overall selectivity to **PC** is about the same when running in isopropanol, THF, Me-BuOH, and Me-THF, and conversion limits the yield to **PC** with these other solvents. However, it seems the product distribution is slightly more favorable when using THF since **1H** is one of the only side products and this should be more advantageous to forming additional amounts of **PC** than products 2 and 3 seen in the reactions with isopropanol. Did the authors run the reaction in THF for longer reaction times to reach full conversion? Would reaching 100% conversion in THF give similar **PC** yields to reactions in IPA and still have **1H** present?

Answer #7, Reviewer #1: Following these suggestions, we have performed the reaction at prolonged time using THF as solvent (8 h). Overall, these results in THF were found very similar to those obtained in isopropanol, displaying ~ 85% **PC** yield. The results have been added in **Table S2** where catalytic results for demethoxylation and hydrogenation of **1G** using THF as solvent at two different reaction times have been listed. It seems that this is a reaction rate issue, which is higher in

isopropanol. This is very likely due to the hydrogen-donor nature of isopropanol compared to THF where the source of H is only the added hydrogen gas. Moreover, the latter solvent is cheaper and is considered greener than THF. Thus, isopropanol was kept as optimal solvent for further optimizations for demethoxylation and hydrogenation of **1G** and the treatment of lignin oil to **PC** diol.

Question #8, Reviewer #1: Table 1/Figure 3. The authors calculated selectivity in terms of moles of propanol syringol/guaiacol converted to **PC**. With the amount of each compound written in terms of mass in Figure 3 and with the conversion being in terms of mass for the rest of the manuscript, this distinction is not very intuitive to determine right away. The authors should include in the table caption for Table 1 that selectivity is defined on a molar basis or cite the equation used in supplement section 1.3.

Answer #8, Reviewer #1: We agree, and according to this suggestion, we have added a footnote regarding the selectivity and yield, which were defined on a molar basis with the equation referred in the table caption for **Table 1** in the revised manuscript.

Question #9, Reviewer #1: Figure 5: The authors note they achieve an overall mass efficiency of 36% from lignin to **PC**, gasoline and jet fuel chemicals. I assume the majority of the mass loss is from removal of oxygen from **HDO** reactions. I think it would be helpful to rephrase/calculate what the efficiency is on a per carbon basis after **RCF**, catalytic funneling, and further upgrading reactions.

Answer #9, Reviewer #1: We appreciate the reviewer's valuable comment. Our earlier calculation of the overall mass efficiency of 36 wt% was in fact based on **C9** units of lignin, considering the removal of oxygen by the **HDO** reaction and bearing in mind theoretical assumptions previously specified in the supporting information – we agree, this was hard to follow. Now, we have re-calculated the lignin utilization efficiency per carbon basis. The detailed calculation is shown in **Supplementary Note 2** and clarifying comments were added to the main text in the revised manuscript.

Overall, the total mass yield of carbon after **RCF**, catalytic funneling and **HDO** is 29.5 %, which is slightly lower than the earlier estimated yield of 36 % based on a **C9** basis.

Question #10, Reviewer #1: The polymerizations were carried out under typical conditions, though the industry and other relevant reports (ref 9, 28, 29 of the main manuscript) of lignin-derived polyesters use Sb_2O_3 as the catalyst and some explanation as to the departure from that compound would be appreciated, or the testing of that catalyst.

Answer #10, Reviewer #1: In this work, we selected $\text{Zn}(\text{OAc})_2$ as it has previously shown excellent reactivity for polyester production, starting from dicarboxylic acid esters and diol co-monomers (Green Chem., 2010, 12, 1704–1706, Polymers, 2017, 12, 693, J. Polym. Environ. 2019, 27, 2167–2181). We agree with the reviewer, that the commonly used Sb_2O_3 should also be tested and compared. Hence, we performed the respective reactions using Sb_2O_3 as catalyst, for the synthesis of poly (**PC**, **TPA**) using **PC** and **DMTA** co-monomers under the previously applied reaction conditions.

However, in the first stage of transesterification performed at 190 °C/ N_2 for 1 h or 2 h or 4 h to make oligomers, the **PC** showed much less reactivity with **TPA**. This has led to the removal of the **PC** and **DMTA** monomers when the second stage polycondensation was carried out under vacuum (1 mPa) at 230 °C. We think therefore that $\text{Zn}(\text{OAc})_2$ was a suitable choice. To describe these experimental findings, we have added a **Table S11** and a short comment in the main text detailing the comparison of reactivity between $\text{Zn}(\text{OAc})_2$ and Sb_2O_3 catalyst for the synthesis of poly (**PC**, **TPA**).

Question #11, Reviewer #1: Molecular weights are comparable along with thermal properties of PET; however, it is notable that these molecular weights are obtained with model compounds in Table 2, Entries 1-8. Entries 9 and 10 with monomer PC isolated from biomass have lower molecular weights and higher dispersities. This is possibly due to the monomer purity being 95%. For polycondensations to achieve higher molecular weight, purity of >99% is typical. It would be gratifying to see if improved monomer purity achieved higher molecular weights with lower dispersity as well as molecular weights after polymerization of PC (produced from RCF) at 3 hours.

Answer #11, Reviewer #1: We thank Reviewer #1 for the excellent comment. Apparently, the display of the various data in the **Table 2** caption was not clear, we apologize for this mistake. In fact, in Entries 9 and 10 we refer to mixtures of diols **PC** (82 %) and **1** (13 %), together occupying 95 % and the remaining 5 % of the mass balance were mono-alcohol impurities **2**, **3** obtained from the catalytic funneling of equimolar model mixtures of **1G/1S** and not from **RCF** oil.

We appreciate the point raised here about the importance of monomer purity for the production of high molecular weight polyesters. In fact, the fractional distillation of the bio-oil produced directly from RCF (process shown in Figure 5), allows the isolation of highly pure monomer mixture consisting of only two components (80% **PC** and 20% diol **1**), where both can participate in the polycondensation. Given the relative structural similarity of **PC** and diol **1**, and their similar boiling point, we were not able to further purify this mixture with the current laboratory equipment available. We believe that such distillative separation would be possible in a larger / industrial setting, should the process be further scaled up. This is an interesting point to investigate. Equally, more studies are needed to determine and clarify in detail the actual influence of the presence of 10-20% diol **1** on the polymer properties.

We have now added a short comment regarding the importance of monomer purity in the synthesis of higher molecular weight and low dispersity polymers and the caption in **Table 2** was modified in the revised manuscript.

Question #12, Reviewer #1: Mechanical and barrier properties associated with PET are extremely relevant for industrial applications. For instance, PEF has not become an attractive alternative due to brittleness and lack of strain-hardening, which limits the application of this material in a pressurized environment like a soda/water bottle. Mechanical property testing and barrier property measurements would be highly impactful for claiming industrial relevance. This is likely an experiment that should be done for this paper.

Answer #12, Reviewer #1: We appreciate the reviewer's valuable indications and we agree that in addition to thermal properties, the mechanical properties and barrier properties of the synthesized thermoplastics are also important to emphasize their industrial relevance. Based on reviewer's comments, the mechanical and barrier properties of poly (**PC/TPA**) and poly (**PC/FDCA**) were investigated and characterized.

Shortly, we believe it is too soon to directly compare to commercial PET since our molecular weight values are not yet optimized.

The results showed, as expected, a difference between our FDCA and TPA based polyester. For the poly (**PC/TPA**), we were able to measure a tensile strength of 13.47 ± 0.50 MPa and Young's modulus is 630.08 ± 12.97 MPa with elongation at break 2.85 ± 0.02 %. As comparison to commercial PET plastics (tensile strength: 55-75 MPa and Young's modulus: 2800-3100), these values are lower, but acceptable, considering that we were not yet able to optimize mechanical properties for our polymers.

Unfortunately we could not measure gas barrier properties for our materials. Our FDCA-based polyester is also brittle, like PEF, but we were able to improve this situation in preliminary experiments by mixing with additional diols. We prefer to not yet show these results, as we would like to perform many more tests on this subject, which will be outside the scope of the current manuscript.

In our case, the number-average molecular weight of the polymers used for determination of mechanical properties, are cca 15 kg/mol, much lower than the commercially bottle-grade PET (24 kg/mol) [Elias HG. *Neue Polymere Werkstoffe Für Die industrielle Anwendung*. 2. Folge. CRC Press; 1986.] and PEF (45-55 kg/mol). [Front. Chem. 2020, 8, 585].

Thus, the molecular weight of our polymers should be improved to make a valid comparison and in order to improve tensile strengths. This should be done in conjunction with further purification of the monomer PC that is used for the making of these polyesters as monomer purity and stoichiometry will further improve Mw and different catalysts and reaction conditions can also be investigated further.

As mentioned, another interesting direction is to use a co-monomer, either ethylene glycol or a longer chain aliphatic diol. This could be integrated in a further biorefinery concept, and can be obtained from the surplus of the cellulose or hemicellulose streams. Other interesting internal plasticisers, bio-based monomers would be longer chain bio-based diacids (e.g. sourced from fatty acids/fatty acid esters), which will reduce possible crystallinity of our material, especially the FDCA based polymers.

We believe, also in view of literature reports, that our monomer that is at the has, at the same time rigid ring as well as more flexible methylene bridges, would be a good candidate for further and relative easy tuning the properties.

Reviewer #2 (Remarks to the Author):

The new strategy producing fully lignocellulose-based PET alternatives with excellent properties and recyclability seems promising. Extensive characterization of products, reaction kinetics and reaction mechanisms have been performed. The experimental results on the production of DMFT and DMTA from the carbohydrate pulp part seem a gap in this work. Process economics is another missing piece. These are the major comments and few more minor comments to be addressed before considering it for publication.

Question #1, Reviewer #2: 1. Improve the experimental results on the production of DMFT and DMTA from the carbohydrate pulp.

Answer #1, Reviewer #2: We highly appreciate the positive evaluation of Reviewer #2 regarding our work and the excellent suggestions. In the previously submitted manuscript we focused experimentally on the lignin fraction which constituted already a large body of data. Since the cellulose conversion pathways are already established and optimized in the literature, these were considered on a theoretical basis.

To address the question of **Reviewer #2**, we have experimentally performed the requested cellulose valorization steps, including RCF of beech wood, catalyst separation and follow up conversion to monomers, and agree that this provides a much more integrated picture. Especially the separation of the cellulose residues from the catalyst after RCF using Cu₂₀-PMO was looked at in detail. This is also in line with the **question #4** of **Reviewer #1**.

To address the comment of **Reviewer#2**, we have first developed a straightforward protocol using sieve fractionation (inspired by *Biotechnol. Biofuels* 2020, 13, 2) to liberate the spent Cu-PMO catalyst from the carbohydrate pulp after **RCF**, whereby we confirmed the removal of catalyst by ICP analysis. Next, the isolated carbohydrate fraction was subjected to three reaction steps (Step 1: Mild enzymatic hydrolysis of cellulose pulp to D-glucose; Step 2: Isomerization and dehydration of D-glucose to **5-HMF**; Step 3: Catalytic oxidation of **5-HMF** to **FDCA**) to result in **FDCA** in a mass yield of 32.7 wt. % on a cellulose basis. This is a good combined yield, directly starting from real carbohydrate pulp obtained from RCF of lignocellulose. In addition, this particular route was chosen due to its practicability, and because we did not have the possibility to explore all the best literature available systems. But these experimental results convincingly show that the amount of **FDCA** produced by our method fits the overall concept perfectly, and the quantities of the cellulose and lignin-derived monomers are already matching.

In the future, the **FDCA** yield can be further optimized by selecting best practices from the literature. For example, from cellulose directly to **5-HMF** (83.3 % yield) as described in *Bioresource Technol.* 2019, 279 84–91 and from **5-HMF** to **FDCA** (99 % yield) in line with *ACS Sustainable Chem. Eng.* 2016, 4, 9, 4752–4761. In this case, it is estimated that **FDCA** will be in surplus and will be considered next to the polymers, as valuable product of this refinery. With 5-HMF already obtained from carbohydrate pulp, the route from **5-HMF** to **TPA** can also be optimized by selecting the best literature data. For example, from **5-HMF** to **DMF** (100 % yield) refers to *Green Chem.* 2014, 16, 1543—1551 and from **DMF** to xylene (refers to *ChemCatChem*, 2017, 9, 398–402) as well as from xylene to **TPA** (93 %) refers to the Amoco process implemented already in industry.

The respective data were added into the Supplementary information, and the TEA calculation has considered both the experimental as well as the literature-based options.

Question #2, Reviewer #2: 2. Add process economics.

Answer #2, Reviewer #2: On the basis of the experimental data, we performed the requested techno-economic assessment (TEA) of the process. The comprehensive evaluation includes the catalytic processing of beech lignocellulose by RCF, followed by the fractionation of the obtained bio-oil and the catalytic processing of the respective fractions to final products. This includes the Raney Nickel mediated catalytic funneling of the monomers to **PC** diol, which is then converted to the respective fully bio-based polyesters; as well as the separation and conversion of the carbohydrate rich residues to **FDCA**.

Overall the process converts beech wood into 1% of gasoline, 1% of jet fuel, 4% of PET and 4 % methanol, 11% of furfural and 10 % of **FDCA** on a mass basis (this being good efficiency with 80% of lignocellulose converted and deoxygenation taken into account). In order to assess this chemical process at such an early stage of development, some basic assumptions have been made also in line with [Science 2020, 367, 1385-1390] and [Perry's Chemical Engineering's Handbook, Section 9, pp. 1-56]. Thus, we estimated fixed operating costs, utility costs and annualized capital cost as a relative share based on raw material costs [React. Chem. Eng., 2021, 6, 225-234] and assumed solvent recovery to reach 98-99% and catalyst recycling.

It is very encouraging, that with the *currently achieved product yields*, and with the assumptions made in line with literature data, the techno-economic evaluation shows a positive balance. More specifically, a 6.4% rate of return can be achieved at 99% solvent recovery. Returns are sensitive to methanol and isopropanol recovery at 96% and 98% respectively. Overall, our analysis indicates that catalyst and solvent costs are the main drivers of operating costs, which is not surprising considering the lab-scale development stage of the process. Furthermore, **FDCA** and furfural represent the most important revenue streams whilst fuels are neglectable in both volume and value. Hence, the profitability of the process is particularly depending on future **FDCA** price assumptions. Since no mature **FDCA** market is yet existing, thus the revenue values estimated in previous papers have been used [Comput. Chem. Eng., 2013, 52, 26-34; Biofuels, Bioprod. Bioref., 15: 1021-1030, Biofuels, Bioprod. Bioref., 13, 1234-1245].

It is clear that up-scaling would focus on reduction of the solvent demand and consumption while optimizing its recovery. Another factor that future optimization may improve, is the total process yield, which is currently at 30 wt%. While this yield is already high considering the well-defined product streams obtained, we still see possibilities for improvement. For example, while the 12 wt. % yield of **PC** is among the best in available literature for a lignin-based polymer building block, this value can be improved by optimizing the catalyst type and flow/vs batch operation of the RCF to maximize **1G/1S** yield. For example, one of the highest yields to **1G/1S** mixture in the literature is 44.8 wt. % [Biotechnol. Biofuels, 2020, 13, 1-10], compared to 24.2 wt. % in this work. Consequently, the **PC** yield could achieve about double the amount currently observed, thereby favorable influencing the techno-economic assessment. In fact, such an increase in yields would enable a profitable (7% return) operation of the process even under the assumption of the lowest possible **FDCA** prices discussed in literature [Biofuels, Bioprod. Bioref., 15, 1021-1030].

Another important aspect is to carefully assess other benefits of bio-based products compared to fossil-based ones, especially in relation to carbon-neutrality and climate benefits. Our process is utilizing a relatively cheap raw material (Beech wood) and targets well defined and already existing products. However, current prices of the substituted fossil-based products are too low considering they are made from rather cheap bulk petrochemicals. However, when assuming emission pricing in the range of 50-100 Euros per ton CO₂ released would add between 2 and 4% to the overall profitability.

We have added a new **Supplementary Note 3** regarding the TEA analysis of our proposed model biorefinery that derives valuable product streams, namely fuels, chemicals and PET mimics from beech wood. Furthermore, several clarifying comments have been inserted into the main text of the revised manuscript, and these are labelled in yellow.

Question #3, Reviewer #2: 3. Step 3 is not shown in the Figure 1.

Answer #3, Reviewer #2: The step 3 was added into **Figure 1b** in the revised manuscript.

Question #4, Reviewer #2: 4. In Figure 5 and Fig S74, hemi(cellulose) content should be 8120 mg instead of 9120 mg.

Answer #4, Reviewer #2: We have performed the respective composition analysis, which showed that the cellulose and hemicellulose content of the carbohydrate rich pulp is 3920 mg (39.2 %) and 1910 mg (19.1 %), respectively. These numbers have been added into the **Figure 5** and **Figure S74** in the revised manuscript and Supplementary information.

Question #5, Reviewer #2: 5. Inconsistency of unit, “bar” vs. “MPa”.

Answer #5, Reviewer #2: The unit MPa has been replaced with bar in **Table 1** in the revised manuscript.

Question #6, Reviewer #2: 6. Do the authors look into what makes Raney Ni more superior than the other 0

Answer #6, Reviewer #2: We attribute the higher catalytic activity of Raney Nickel to the fact that it is a highly active transfer hydrogenation catalyst. This makes for a facile hydrogen abstraction from the H-donor isopropanol, as it has earlier been demonstrated in excellent works of Rinaldi (Energ. Environ. Sci. 2012, 5, 8244–8260, ACS Catal., 2017, 7, 2437–2445).

It is also to be noted, that the other noble metal catalysts that were tested, possess a higher affinity for aromatic ring reduction compared to Raney Nickel, while in this particular case, facile demethoxylation over aromatic ring hydrogenation is desired, since demethoxylation starting from the saturated ring is much slower, as also showed in our mechanistic studies (See Fig. 2E). One of us has also recently summarized the advantages of Raney nickel for such hydrodeoxygenation reactions, for example the hydrodeoxygenation of guaiacol to cyclohexanol, in a recent review article (ACS Catal. 2021, 11, 10508–10536).

In agreement with **Reviewer #2** that these points require more discussion, we have inserted a **Supplementary Note 4** to the Supporting information and respective comments to the main text, labelled in yellow.

Question #7, Reviewer #2: 7. Figure 2 labels and axes labels are hard to read.

Answer #7, Reviewer #2: We apologize that **Figure 2** has not been formatted properly. The **Figure 2** has been reformatted and added into the revised manuscript.

Question #8, Reviewer #2: 8. Optimized temperature was said to be 140 °C. However, Figure 2D only shows results up to 120 °C

Answer #8, Reviewer #2: This discrepancy was due to the different reactivity of **1G** and **1S**. While for the demethoxylation and hydrogenation of **1G** to **PC** (85%), optimum temperature and time were found to be 120 °C and 2h, applying these same reaction conditions to **1S** led to lower **PC** yield (63 %). Further optimization involving **1S** found that 140 °C needs to be used. Such optimization runs using pure model compounds were necessary, as the final goal was to achieve the best yield of **PC** in the catalytic funneling of **1S/1G** in lignin oil.

To better address the question of the **reviewer #2**, we have added a **Table S5** in the manuscript.

Question #9, Reviewer #2: 9. Typo in conclusion, “PTA” should be “TPA”.

Answer #9, Reviewer #2: The typo **PTA** has been corrected as **TPA** according to this thoughtful suggestion.

Reviewer #3 (Remarks to the Author):

Overall the authors have presented a novel biorefinery system that converts wood to fuels and useful packaging plastics. The technique of reductive catalytic fractionation (RCF) was adapted very well to the deconstruction of woody biomass. There is considerable novelty and value in the (relatively) high conversion of lignin to the aliphatic diol, PC (hydroxypropylcyclohexanol). The creation PET mimics with PC was clearly demonstrated as such polymers match or slightly excel the T_g of PET. In turn, depolymerization (via methanolysis) of the PET mimics was demonstrated and thereby, shown to be a viable step in the circular recycling of these materials. I recommend publication in Nature Communications after attention is paid to the points made above.

We are very pleased by the positive evaluation of our work and great suggestions by **Reviewer #3** which we have fully addressed below and in the manuscript.

Some grammatical issues to be addressed include:

Question #1, Reviewer #3: Page 3: “using Raney nickel led to” (not “using Raney nickel lead to”)

Answer #1, Reviewer #3: The typo ‘lead to’ has been corrected as ‘led to’ in the revised manuscript.

Question #2, Reviewer #3: Page 3: “As a comparison,” (add “a”)

Answer #2, Reviewer #3: The typo ‘As comparison’ has been replaced with ‘as a comparison’ in the revised manuscript.

Question #3, Reviewer #3: Figure 1a: The on syringic acid PEVA structure drawn is based and probably should be PESA. The T_g of 82 also suggests this is the syringic acid polymer. PEVA probably applies to the vanillic acid variant, but it has a lower T_g . There is something under the wood graphic that probably was meant to be in front of the wood graphic.

Answer #3, Reviewer #3: We appreciate the reviewer’s valuable comment and apologize for this mistake. Indeed, the paper (*Macromol. Rapid. Comm.* 2011, 32, 1386-1392), describes **PEVA** as abbreviation of polyethylene vanillate, which has a T_g of 55 °C, while **PESA** described in (*Green Chem.* 2017,19, 1877-1888) for abbreviation of polyethylene syringate with higher T_g of 82 °C. The abbreviation **PEVA** has been replaced with **PESA** in Figure 1b in the revised manuscript.

Question #4, Reviewer #3: Page 9: “in applications similar to those of PET or as” (not “in applications similar to the application are of PET or as”)

Answer #4, Reviewer #3: The mistake ‘in applications similar to the application are of PET or as’ has been replaced with ‘in applications similar to those of PET or as’ in the revised manuscript.

Question #5, Reviewer #3: Page 9: “glass transition temperatures (T_g) between” (add “s”)

Answer #5, Reviewer #3: The typo ‘glass transition temperature’ has been changed to ‘glass transition temperatures’ in the revised manuscript.

Question #6, Reviewer #3: The PC yield is given as 85% from 1G/1S in Figure 1. But the text claims “56.4% efficiency,” which I assume is the yield from lignin. But the yields of PC in Figure 3 are 15% maximum. Yet another “estimated” value of 36.2% “utilization efficiency” is mentioned in the conclusions. The overall yield from the lignin starting material should be better described in the introduction—perhaps the mass yield from the biomass wood starting material. A 15% production of one molecule from lignin seems to be a great result—much higher than the state of the art alternative

procurements, such as vanillin from spruce (perhaps 1% from wood itself). But has the PC been separated and purified? What is the real percent yield of pure, isolated PC?

Answer #6, Reviewer #3: We apologize for not clarifying these numbers well, although all of these numbers have been defined and explained in the previous manuscript, we agree that it was not easy to follow. In fact, the 85 % yield of PC was obtained by the demethoxylation and hydrogenation of the model compounds mixture **1G/1S**, while 56.4 % efficiency was achieved based on the assumption that lignin is composed of two types of building units, namely, 4-propanolguaiacol/syringol. And another estimated value of 36.2 % “utilization efficiency” was given based on **C9** balance (4-propylcyclohexane) after O was completely removed.

The 15 wt. % PC yield mentioned, obtained by the processing of bio-oil, was quantified by GC-FID measurement using internal standard. We also indicated the isolated yield of 11.7 % for PC (based on lignin content) in the Supplementary information 2.5.

In order to more coherently display the efficiency, and also in line with the question of [**Reviewer #1, Question # 9**], we now made the following changes to the manuscript:

- The theoretical estimated value of 56.4 % in the abstract was removed and 11.7 wt % PC isolated yield was reintroduced.
- The estimated value of 36.2 % was removed and a more accurate carbon utilization value of 29.5 % was reintroduced in **Fig. 1b**. Furthermore, a sample calculation was displayed in the revised **Supplementary Note 2**.

Question #7, Reviewer #3: Figure 1b: And the left “OH” group of PC should be “HO” as drawn.

Answer #7, Reviewer #3: This has been corrected in **Figure 1**.

Question #8, Reviewer #3: Figure 1b: Molecular weights are provided with too many significant figures. Typically, the tens and one positions are both zero, with most instruments. Also, T_g values usually have no decimal places. Conventionally, one would describe a copolymer as poly (PC/TPA) and not Poly (PC, TPA).

Answer #8, Reviewer #3: Based on your suggestions, molecular weights, T_g, and polymer names were all corrected in the revised manuscript.

Question #9, Reviewer #3: Page 8: “dyad” (not “diad”, according to the dictionary and experienced polymer chemists)

Answer #9, Reviewer #3: We are thankful for all the suggestions of the expert referee also regarding aspects of polymer chemistry. The typo ‘diad’ has been corrected as dyad in the revised manuscript.

Question #10, Reviewer #3: Page 9: The T_g of 51°C for the trans polymer seems rather low compared to the other stereochemical variants. It is also has the lowest molecular weight among the PC/FDA and PC/DMT variants. Perhaps the Mw is too low to provide a realistic T_g value here and a comment could be added to warn this.

Answer #10, Reviewer #3: We are grateful for the reviewer’s valuable comment. A short note regarding to ‘The lower value of T_g for poly (PC_{trans}/TPA), compared to poly (PC_{trans}/TPA) could be attribute to low M_w value (21.2 kg·mol⁻¹ versus 16.8 kg·mol⁻¹)’ has been added into the manuscript.

Question #11, Reviewer #3: Page 9: Poly (PC/1/FDCA) is lauded as being highly biobased and having a high T_g value near 74 °C. FDCA, while biobased, is still very expensive to produce. Moreover, poly(ethylene glycol/FDCA) (polyethylene furanoate, PEF), the long-promised commercial product

from FDCA has an even higher Tg near 86 °C—and uses a currently available and inexpensive bio-based diol, ethylene glycol. So while there are certainly some commercial possibilities with PC, it performs no better than ethylene glycol. One advantage that could be mentioned is that EG derives from a food source, Brazilian sugar, but PC derives from a non-food source, lignin.

Answer #11, Reviewer #3: We are grateful for the excellent comments - we agree with these. In fact our main goal here was to provide a novel integrated biorefinery concept to produce several products, including an alternative fully bio-based PET analogue, which can be obtained entirely from lignocellulose, a non-edible desirable raw material. Of course, there are several hurdles this field has to overcome in order to implement lignocellulose-biorefinery based products, and we believe that our solution provided here, can represent one possible alternative, given its relative practicality. Note that RCF in general is already being scaled by others, our follow-up purification and catalytic funneling process is relatively simple and using an industrially compatible and recyclable catalyst and provides the desired product in high yields. Surely, it is too early to say whether our **PC**-based fully bio-based polyesters will have any commercial application, though other excellent scientific papers producing PET analogues from renewables face numerous hurdles, such as the use of specialized catalysts, expensive monomer purifications or non-practical monomer yields. It is also interesting to look into the development of PEF from conception towards practical application. One recognizes the large amount of work that goes into optimizing process as well as polymer properties, ultimately reaching practical applications. As we detailed in our answer to **Reviewer #2**, initial techno-economic analysis shows feasibility and there are several aspects related to the process as well as the polymer properties that can be optimized in future work.

We have now included a respective discussion into the conclusions section that puts the work into the proper context and perspective and also highlights future possibilities.

REVIEWER COMMENTS

Reviewer #1 (Remarks to the Author):

The authors have fully addressed my comments. It is clear that they have done an Herculean amount of work on the revision of this manuscript, and I suggest that it proceed to publication from here.

Reviewer #2 (Remarks to the Author):

The authors have revised a few of the comments but some of them have not been properly addressed. These are rather minor points but necessary for proper credit, reproducibility, etc.

- One of the comments was to improve the experimental results on the production of DMFD and DMTA from the carbohydrate pulp. The authors claimed the cellulose conversion pathways are already established and optimized in prior work. The authors' lignin work (this manuscript) consists of a large body of data. In the revised manuscript, the authors did perform cellulose valorization to FDCA. Although not optimized, the paper now provides a much more integrated picture. It would still be good to mention briefly and cite that the cellulose conversion pathways are already established and optimized.

- Also, include references related to DMFD and DMTA synthesis.

- Figure 2E and 2F are still blurry.

- Another comment was to include process economics, which it is in the revised manuscript. The authors said that it is still difficult to compete with fossil-based PET or fuels (cost and process).

- o It would be good to include actual numbers on the potential selling price of PC/TPA and PC/FDCA and predicted CO₂ emissions as well as comparison with fossil-based processes/products. Even if the numbers are not competitive.

- o The authors mentioned "table 3" on supplementary note 2, however table 3 does not exist.

- o No methods and software used for the TEA analysis are reported. Include diagrams of the simulation in the supplementary document to enable reproducibility of the work.

Reviewer #3 (Remarks to the Author):

The revisions made suitably address all of my original comments and concerns.

Responses to the points raised by the Reviewers

Reviewer #1:

The authors have fully addressed my comments. It is clear that they have done an Herculean amount of work on the revision of this manuscript, and I suggest that it proceed to publication from here.

Reviewer #3:

The revisions made suitably address all of my original comments and concerns.

Answer to Reviewer #1 and #3

We appreciate the acknowledgement of our efforts to thoroughly address the reviewer's comments. We believe to have carefully considered all the comments and hope that the manuscript is now suitable for publication.

Reviewer #2 (Remarks to the Author):

The authors have revised a few of the comments but some of them have not been properly addressed. These are rather minor points but necessary for proper credit, reproducibility, etc.

Question #1, Reviewer #2: One of the comments was to improve the experimental results on the production of DMFD and DMTA from the carbohydrate pulp. The authors claimed the cellulose conversion pathways are already established and optimized in prior work. The authors' lignin work (this manuscript) consists of a large body of data. In the revised manuscript, the authors did perform cellulose valorization to FDCA. Although not optimized, the paper now provides a much more integrated picture. It would still be good to mention briefly and cite that the cellulose conversion pathways are already established and optimized.

Answer #1, Reviewer #2: We appreciate the reviewer's valuable comments for the further improvements of the manuscript.

The production of FDCA from cellulose has been investigated widely. For example, the conversion of cellulose directly to 5-HMF (83 % yield) is described in *Bioresource Technol.* **2019**, *279*, 84-91 and the respective 5-HMF to FDCA (99 %) conversion in *ACS Sustainable Chem. Eng.* **2016**, *4*, 4752-4761.

We have added a respective comment regarding the optimized cellulose conversion pathway to FDCA into the revised manuscript and **Supplemental Note 8**, with the corresponding references cited.

Question #2, Reviewer #2: • Also, include references related to DMFD and DMTA synthesis.

Answer #2, Reviewer #2: Further DMFD formation starting from FDCA can be inferred with a high yield (up to 99 %) as shown in *Patent*, **CN111072511**, **2018**. Furthermore, we have selected and listed the best literature regarding the catalytic conversion of 5-HMF to DMTA in Supplementary section 1.6. For example, the transformation from 5-HMF to DMF (100 %) refers to *Green Chem.* **2014**, *16*, 1543-1551, from DMF to xylene (0.97 %) refers to *ChemCatChem*, **2017**, *9*, 398-402, from xylene to TPA (0.93 %) refers to the Amoco process (*Chem. Soc. Rev.* **2020**, *49*, 5704-5771) implemented already in industry; and the reaction from TPA to DMTA (99%) refers to *Chem. Eur. J.* **2018**, *24*, 2360-2364.

We have added respective comments and corresponding references for the synthesis of DMFD into the supplementary information (**Supplementary Note 8**). The optimized route for the synthesis of DMTA was already present in the Supplementary information (**Supplementary Note 1**).

Question #3, Reviewer #2: Figure 2E and 2F are still blurry.

Answer #3, Reviewer #2: Thank you for your indication. We have added a higher resolution Figure 2E and 2F in the revised manuscript.

Question #4, Reviewer #2: Another comment was to include process economics, which it is in the revised manuscript. The authors said that it is still difficult to compete with fossil-based PET or fuels (cost and process). It would be good to include actual numbers on the potential selling price of PC/TPA and PC/FDCA and predicted CO₂ emissions as well as comparison with fossil-based processes/products. Even if the numbers are not competitive.

Answer #4, Reviewer #2:

Thank you for the valuable comment. More discussion with regard these points have been added. In Supplementary Table 13 the estimated revenues were already shown and now **Supplementary Table 14** has been added, which summarizes CO₂ emission values.

With regard to selling prizes, in our preliminary calculation for PC/TPA we used the same prices as PET, which were assumed to reach 1500 USD/ton based on recent industry reports. However, current reports would indicate even higher prices, like 1600 to 1700 Euro/ton for recycled PET (<https://packagingeurope.com/news/rpet-prices-reach-record-high-across-europe/6953.article>).

In this simple preliminary model used, FDCA produced in surplus in our process, and is regarded a value-added chemicals, makes most of the contributions to revenues. Since no mature FDCA market yet exists, all prices of FDCA proposed in reported literatures are educated guesses. FDCA prices used were referred to *Comput. Chem. Eng.* **2013**, *52*, 26-34. (2458-3885 USD/ton), *Biofuel Bioprod. Bior.* **2021**, *15*, 1021-1030 (1200-2000 USD/ton) and *Biofuel Bioprod. Bior.* **2019**, *13*, 1234-1245 (1800 USD/ton). While our initial process requires a rather high FDCA price (at least 2800 USD/ton) to prove profitable, an optimized process with higher yields could perform profitable already at 1000 USD/ton.

With regards to CO₂ emissions, for gasoline, PET, jet fuel and methanol, we considered CO₂ emissions based on ECOINVENT data (CO₂ avoided kg/kg of product) being 0.602, 2.938, 0.447 and 0.361 respectively – these data were displayed in Supplementary Table 14.

Question #5, Reviewer #2: The authors mentioned "table 3" on supplementary note 2, however table 3 does not exist.

Answer #5, Reviewer #2: We apologize for this mistake. The Table 3 has been corrected as supplementary Table 13 in **Supplementary Note 8**.

Question #6, Reviewer #2: No methods and software used for the TEA analysis are reported. Include diagrams of the simulation in the supplementary document to enable reproducibility of the work.

Answer #6, Reviewer #2: We performed the process economics analysis based on currently obtainable experimental data and available product price data cited from the literatures and commercial webpages. At the current stage of development this assessment is considered as a prospective, order of magnitude estimate, typical for TRL 3 to 4 (*Biofuel Bioprod. Bior.* **2014**, *8*, 635-644). The calculations were performed in Excel, and this approach has been proven successful in recent literature, see: *Sustainability* **2021**, *13*, 2063, *Clean Technol. Envir.* **2016**, *18*, 1849-1862 and *ACS Sustainable Chem. Eng.* **2021** *9*, 3428-3438.

The figure above shows the results of 60 different scenarios when varying raw material costs, chemical costs, recovery rates, CO₂ prices, product prices, catalyst costs, always considering two different yield levels and with and without CO₂- taxation. The results turn negative in case of lower recovery and lower furfural prices. If these scenarios can be avoided we can see that most runs (scenarios) provide rates of return between 5 and 13 %.

This explanation has been added within Supplementary note 8, and the graph as Supplementary Figure 100.